# Liver type 1 innate lymphoid cells lacking IL-7 receptor are a native killer cell subset fostered by parenchymal niches

Takuma Asahi[1,2], Shinya Abe[1,2], Guangwei Cui[1], Akihiro Shimba[1,3], Tsukasa Nabekura[4,5,6], Hitoshi Miyachi[7], Satsuki Kitano[7], Keizo Ohira[1,8], Johannes M Dijkstra[9], Masaki Miyazaki[10], Akira Shibuya[4,5,6], Hiroshi Ohno[11], Koichi Ikuta[1]*

[1]Laboratory of Immune Regulation, Department of Virus Research, Institute for Life and Medical Sciences, Kyoto University, Kyoto, Japan; [2]Graduate School of Medicine, Kyoto University, Kyoto, Japan; [3]Department of Human Health Sciences, Graduate School of Medicine, Kyoto University, Kyoto, Japan; [4]Life Science Center for Survival Dynamics, Tsukuba Advanced Research Alliance (TARA), University of Tsukuba, Tsukuba, Japan; [5]Department of Immunology, Faculty of Medicine, University of Tsukuba, Tsukuba, Japan; [6]R&D Center for Innovative Drug Discovery, University of Tsukuba, Tsukuba, Japan; [7]Reproductive Engineering Team, Institute for Life and Medical Sciences, Kyoto University, Kyoto, Japan; [8]Graduate School of Biostudies, Kyoto University, Kyoto, Japan; [9]Center for Medical Science, Fujita Health University, Aichi, Japan; [10]Laboratory of Immunology, Institute for Life and Medical Sciences, Kyoto University, Kyoto, Japan; [11]RIKEN Center for Integrative Medical Sciences, Yokohama, Japan

*For correspondence:
ikuta.koichi.6c@kyoto-u.ac.jp

Competing interest: The authors declare that no competing interests exist.

**Abstract** Group 1 innate lymphoid cells (G1-ILCs), including circulating natural killer (NK) cells and tissue-resident type 1 ILCs (ILC1s), are innate immune sentinels critical for responses against infection and cancer. In contrast to relatively uniform NK cells through the body, diverse ILC1 subsets have been characterized across and within tissues in mice, but their developmental and functional heterogeneity remain unsolved. Here, using multimodal in vivo approaches including fate-mapping and targeting of the interleukin 15 (IL-15)-producing microenvironment, we demonstrate that liver parenchymal niches support the development of a cytotoxic ILC1 subset lacking IL-7 receptor (7 R⁻ ILC1s). During ontogeny, fetal liver (FL) G1-ILCs arise perivascularly and then differentiate into 7 R⁻ ILC1s within sinusoids. Hepatocyte-derived IL-15 supports parenchymal development of FL G1-ILCs to maintain adult pool of 7 R⁻ ILC1s. IL-7R⁺ (7R⁺) ILC1s in the liver, candidate precursors for 7 R⁻ ILC1s, are not essential for 7 R⁻ ILC1 development in physiological conditions. Functionally, 7 R⁻ ILC1s exhibit killing activity at steady state through granzyme B expression, which is underpinned by constitutive mTOR activity, unlike NK cells with exogenous stimulation-dependent cytotoxicity. Our study reveals the unique ontogeny and functions of liver-specific ILC1s, providing a detailed interpretation of ILC1 heterogeneity.

## Editor's evaluation

This study provides important insights into the developmental process and functional heterogeneity of liver ILC1s, especially how IL-7R+ and IL-7R- ILC1s are generated. The authors present compelling evidence on the dependence of ILC1s on IL-7R- precursor and their reliance on IL-15 to develop cytotoxic functions. The work will be of broad interest to immunologists and liver biologists.

## Introduction

Group 1 innate lymphoid cells (G1-ILCs) are innate immune cells contributing to surveillance of intracellular infections and tumors. G1-ILCs comprise two subtypes: natural killer (NK) cells and type 1 ILCs (ILC1s), that share fundamental features such as NK1.1$^+$NKp46$^+$ phenotype, expression of T-bet, and IFN-γ production (*Jacquelot et al., 2022*; *Stokic-Trtica et al., 2020*; *Vivier et al., 2018*). In contrast, mouse ILC1s can be distinguished from NK cells by their CD49a$^+$CD49b$^-$ phenotype, strict tissue-residency, and Eomes-independence (*Daussy et al., 2014*; *Gasteiger et al., 2015*; *Peng et al., 2013*; *Sojka et al., 2014*). Additionally, ILC1s and NK cells are developmentally different in general, as confirmed by the fact that ILC progenitors (ILCPs) expressing PLZF and/or PD-1 as well as liver-resident Lin$^-$Sca-1$^+$Mac-1$^+$ (LSM) and Lin$^-$CD49a$^+$CD122$^+$ ILC1 precursors preferentially differentiate into ILC1s more than NK cells (*Bai et al., 2021*; *Constantinides et al., 2015*; *Constantinides et al., 2014*; *Yu et al., 2016*).

Functional differences between NK cells and ILC1s have also been recognized, though some confusion remains, particularly in cytotoxicity. In mice, NK cells are traditionally considered as more cytotoxic than ILC1s (*Vivier et al., 2018*), although this view has been questioned recently (*Dadi et al., 2016*; *Kansler et al., 2022*; *Krabbendam et al., 2021*; *Nixon et al., 2022*; *Yomogida et al., 2021*). Indeed, murine NK cells show low expression of cytotoxic molecules and only minimal cytotoxicity in their steady state (*Fehniger et al., 2007*). The cytotoxicity of NK cells requires the mTOR-dependent metabolic reprogramming mediated by cytokine signaling such as IL-15 or by NK receptor engagement (*Marçais et al., 2014*; *Nandagopal et al., 2014*), but whether such cytotoxic machinery also exists in ILC1s is unclear. By contrast, ILC1s can immediately respond and produce IFN-γ during liver injury (*Nabekura et al., 2020*) and virus infection (*Weizman et al., 2017*), highlighting the unique roles of ILC1s in the ignition of type 1 immunity in tissues. Thus, examining the development, function, and heterogeneity of ILC1s could lead to further understanding of local immune regulation and novel therapeutic strategies.

Recent high-resolution analysis has uncovered ILC1 heterogeneity and development. Liver ILC1s are separated into IL-7R-negative (7 R$^-$) and -positive (7R$^+$) subsets (*Friedrich et al., 2021*; *Sparano et al., 2022*; *Yomogida et al., 2021*). G1-ILCs in the fetal liver (FL) are identified as precursors of ILC1s (*Chen et al., 2022*; *Sparano et al., 2022*), especially of Ly-49E$^+$ ILC1s that are included in 7 R$^-$ ILC1s (*Chen et al., 2022*). In addition, 7R$^+$ ILC1s in the liver, salivary glands (SG), and small intestines can give rise to 7 R$^-$ ILC1s in response to cytokines and inflammations (*Friedrich et al., 2021*), suggesting that 7R$^+$ ILC1s are also potential precursors for 7 R$^-$ ILC1s. However, several reports have suggested different models. In ILC1-related inflammation models reported so far, including contact hypersensitivity, MCMV infection, and liver injury, ILC1s with high cytokine receptors (IL-7R, IL-18R, and/or CD25) are induced and accumulate in the liver (*Nabekura et al., 2020*; *Wang et al., 2018*; *Weizman et al., 2019*). Furthermore, an organ-wide single-cell RNA sequencing (scRNA-seq) analysis reveals that liver 7 R$^-$ ILC1s represent a unique population distinct from SG and small intestines (*McFarland et al., 2021*), suggesting the absence of universal differentiation programs of ILC1s conserved across tissues. Thus, concepts of ILC1 heterogeneity and development are still controversial.

Additionally, environmental factors regulating ILC1 development are poorly understood. Accumulating evidence has shown that development and maintenance of ILCs are strictly associated with their resident tissue microenvironment, called niche (*Ikuta et al., 2021*; *McFarland and Colonna, 2020*; *Murphy et al., 2022*). G1-ILC homeostasis heavily depends on interleukin 15 (IL-15), that is transpresented from hematopoietic and stromal cells as an IL-15/IL-15Rα complex to locally promote the development, survival, and proliferation of memory CD8 T cells, NKT cells, and G1-ILCs in various tissues (*Ikuta et al., 2021*; *Klose et al., 2014*; *Lodolce et al., 1998*). However, how tissue environment regulates ILC1 homeostasis and whether specific niches control the formation of ILC1 heterogeneity are yet to be characterized.

Based on fate-mapping, transfer studies, and targeting of the IL-15-producing microenvironment, we have addressed the developmental processes of heterogenous ILC1 subsets. Adult liver (AL) 7R$^+$ ILC1s are not converted to 7 R$^-$ ILC1s in vivo and RORα deficiency results in selective reduction of 7R$^+$ ILC1s, suggesting that 7R$^+$ ILC1s are not necessary for the development of 7 R$^-$ ILC1s. FL G1-ILCs originate from perivascular sites of the liver and then infiltrate into sinusoids to give rise to AL 7 R$^-$ ILC1s. Hepatocyte-derived IL-15 supports FL G1-ILCs development in parenchyma, thereby maintaining mature 7 R$^-$ ILC1s in sinusoids. Functionally, 7 R$^-$ ILC1s exert inflammation-independent

cytotoxicity through granzyme B expression, which is underpinned by their tonic mTOR activity. Our findings reveal that 7 R⁻ ILC1s represent an ILC subset with unique developmental processes and unconventional native cytotoxicity distinct from NK cells and 7R⁺ ILC1s.

## Results

### Fetal and adult liver contain bona fide ILC1s lacking IL-7R

To characterize and make the relationships among fetal and adult G1-ILCs clear, we first assessed their expression of NK- and ILC-signature molecules. FL G1-ILCs, adult tissue ILC1s, and NK cells were identified as CD49a⁺CD49b$^{int}$, CD49a⁺CD49b$^{lo}$, and CD49a⁻CD49b⁺ populations within G1-ILCs, respectively (*Figure 1A*). FL G1-ILCs highly expressed CXCR6, TRAIL, and CD200R, resembling adult tissue ILC1s, though they completely lacked IL-7R expression (*Figure 1B and C*). AL contains both IL-7R-negative (7 R⁻) and -positive (7R⁺) ILC1s (*Figure 1C*), as reported previously (*Friedrich et al., 2021*; *Sparano et al., 2022*; *Yomogida et al., 2021*). In contrast, CD49a⁺CD49b$^{lo}$ ILC1s in bone marrow (BM), corresponding to previously reported immature ILC1s (iILC1s) that have ability to give rise to liver ILC1s (*Klose et al., 2014*), were mostly IL-7R⁺, similar to other tissues including the spleen, mesenteric lymph nodes, peritoneal cavity, and small intestines (*Figure 1C and D*). Despite the differential IL-7R expression, the frequencies and numbers of AL 7 R⁻ and 7R⁺ ILC1s were not reduced in $Il7^{-/-}$ mice (*Figure 1—figure supplement 1A and B*), consistent with the basic property of G1-ILCs of being IL-7-independent (*Klose et al., 2014*; *Robinette et al., 2017*).

We further assessed the relevance among G1-ILC subsets by bulk RNA sequencing (RNA-seq). All analyzed G1-ILC populations expressed *Tbx21* (T-bet) but not *Rorc* (RORγt), suggesting the lack of ILC3 contamination (*Figure 1—figure supplement 1C*). Consistent with surface phenotype, FL G1-ILCs and AL 7 R⁻ ILC1s as well as BM iILC1s and AL 7R⁺ ILC1s shared high expression of ILC1 signature genes (*P2rx7*, *Zfp683*, and *Cd3g*) and low expression of NK cell signature genes (*Sell*, *Klra4*, and *Klra8*) (*Figure 1E*). By contrast, 7R⁺ ILC1s and BM iILC1s showed higher expression of genes related to cytokine responses (*Il7r*, *Il2ra*, *Icos*, and *Kit*) compared to other G1-ILC subsets (*Figure 1—figure supplement 1D*). Conversely, FL G1-ILCs and 7 R⁻ ILC1s poorly expressed cytokine receptor-related genes (*Il7r*, *Il18r1*, and *Il18rap*) (*Figure 1E*) and exhibited less cytokine-responsive characters (*Figure 1—figure supplement 1E*). Although FL G1-ILCs were unique in terms of their high proliferative status (*Figure 1—figure supplement 1F and G*), principal component analysis (PCA) revealed that the overall transcriptional state of FL G1-ILCs was close to 7 R⁻ ILC1s (*Figure 1F*). 7R⁺ ILC1s located rather close to BM iILC1s in PCA, suggesting their cross-tissue similarity. These results show the transcriptional resemblance between FL G1-ILCs and AL 7 R⁻ ILC1s or between BM iILC1s and AL 7R⁺ ILC1s.

Given the ILC1-like transcriptional programs and T-bet⁺Eomes⁻ phenotype of FL G1-ILCs and AL 7 R⁻ ILC1s (*Figure 1G*), they were considered as ILC1s but not NK cells. To precisely verify their lineage, we performed fate-mapping of ILCPs by using PLZF$^{GFPcre/+}$ Rosa26-YFP reporter (PLZF-fm) mice (*Constantinides et al., 2014*). In these mice, ILCP progenies including ILC1s, ILC2s, and ILC3s rather than LTi or NK cells are preferentially labeled by YFP, though a certain ratio of blood cells expresses YFP due to the pre-hematopoietic PLZF expression, as reported previously (*Constantinides et al., 2014*). To remove this background YFP labelling, we sorted YFP⁻Lin⁻Sca1⁺c-Kit⁺ (YFP⁻ LSK) cells from BM of PLZF-fm mice and transferred them into irradiated WT hosts. In these chimeric mice, AL 7 R⁻ and 7R⁺ ILC1s as well as other adult tissue ILC1s prominently expressed YFP (*Figure 1H and I*). A similar trend was observed when using straight PLZF-fm mice or chimeric mice reconstituted with FL YFP⁻ LSK cells from PLZF-fm mice (*Figure 1—figure supplement 1H and I*). Furthermore, FL G1-ILCs mostly expressed YFP in straight PLZF-fm mice (*Figure 1J*), in line with a previous fate-mapping study of neonatal liver G1-ILCs (*Constantinides et al., 2015*). Thus, these results indicate that *bona fide* ILC1s lacking IL-7R are enriched in FL and AL.

### 7R⁺ ILC1s minimally contribute to the development of 7R⁻ ILC1s

A previous study reported that 7R⁺ ILC1s behaved as the precursors of 7 R⁻ ILC1s when cultured in vitro or transferred into lymphopenic mice (*Friedrich et al., 2021*). However, in the liver, while 7R⁺ ILC1s were nearly absent in infants and accumulated with age, 7 R⁻ ILC1s were predominant in young mice, decreased with age, and eventually depleted (*Figure 2A–C*). These observations suggest that

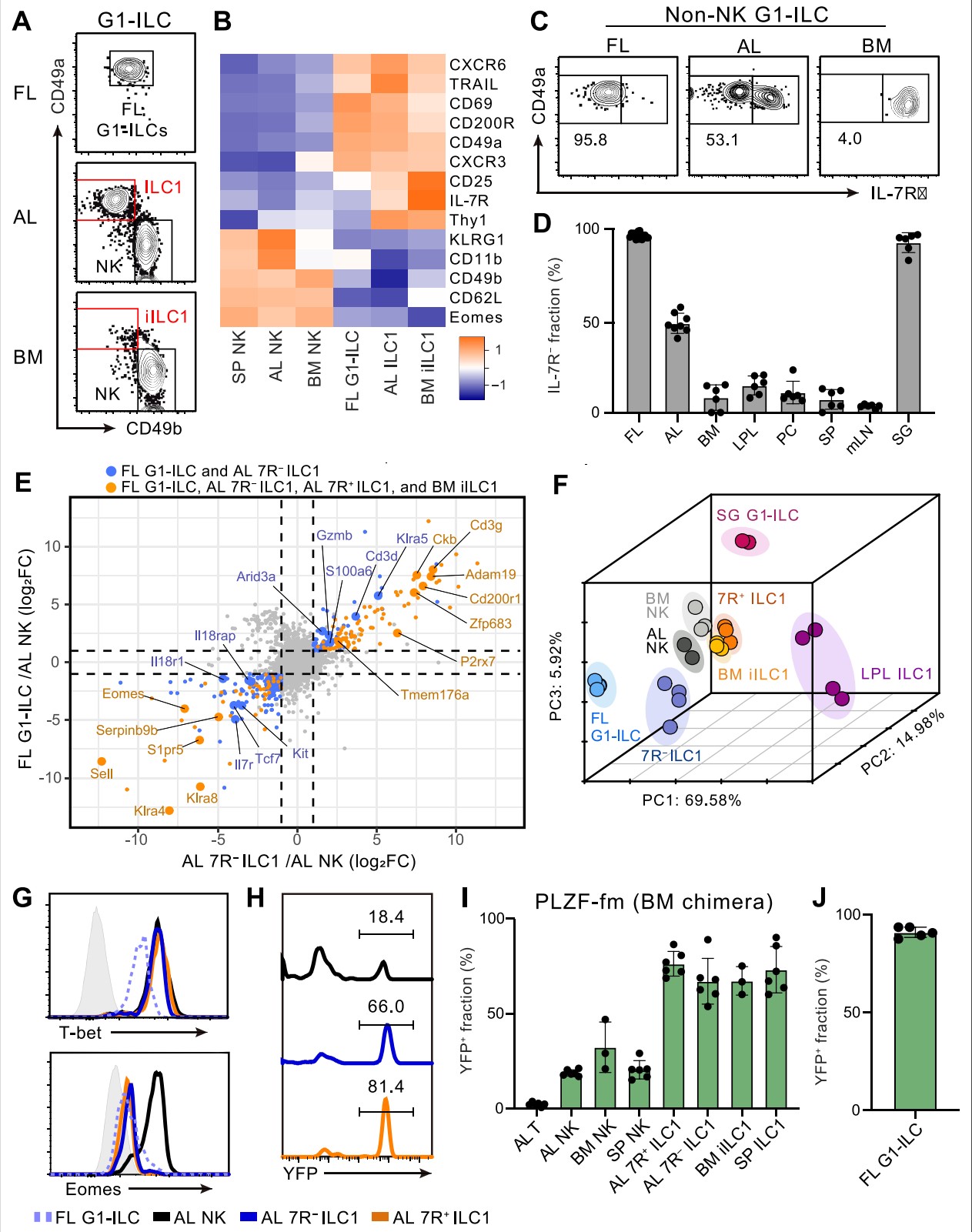

**Figure 1.** Fetal and adult liver contain bona fide ILC1s lacking IL-7R. (**A**) Gating strategy of subpopulations of G1-ILCs (CD3⁻NK1.1⁺NKp46⁺) in the E18.5 fetal liver (FL), adult liver (AL), and bone marrow (BM). iILC1, immature ILC1. Data represent three independent experiments (FL, n=15; AL, n=8; BM, n=6). (**B**) Heatmap representing log2 transformed mean fluorescence intensity (MFI) of indicated protein expression normalized by z-score transformations (n=3 for all subsets). SP, spleen. (**C**) Expression of IL-7Rα on G1-ILCs except for CD49a⁻CD49b⁺ NK cells (non-NK G1-ILCs) in FL, AL,

*Figure 1 continued on next page*

*Figure 1 continued*

and BM. Data represent three independent experiments (FL, n=15; AL, n=8; BM, n=6). (**D**) The percentages of IL-7R⁻ fractions in non-NK G1-ILCs in the indicated tissues. LPL, small intestinal lamina propria lymphocytes; PC, peritoneal cavity; mLN, mesenteric lymph node; SG, salivary gland. Data are pooled from three independent experiments (FL, n=15; AL, n=8; BM, n=6; LPL, n=6; PC, n=6; SP, n=6; mLN, n=6; SG, n=6). (**E**) Scatter plot showing relative gene expression of FL G1-ILCs and AL 7 R⁻ ILC1s compared to AL NK cells in RNA-seq. Genes differentially expressed by FL G1-ILCs, AL 7 R⁻ ILC1s, AL 7R⁺ ILC1s, and BM iILC1s (orange) or only by FL G1-ILCs and AL 7 R⁻ ILC1s (blue) compared to AL NK cells are highlighted. FC, fold change. (**F**) First three principal components in PCA of top 3,000 variant genes. (**G**) Expression of T-bet (upper) and Eomes (lower) on FL G1-ILCs as well as NK cells, 7 R⁻ ILC1s, and 7R⁺ ILC1s in AL. Shaded histograms (grey) indicate isotype controls. Data represent two independent experiments. (**H and I**) Fate-mapping analysis of adult chimeric mice reconstituted with BM YFP⁻Lin⁻Sca1⁺c-Kit⁺ (LSK) cells from PLZF^GFPcre/+ Rosa26-YFP (PLZF-fm) mice. Representative histograms of YFP expression (**H**) and the percentages of YFP⁺ cells in indicated cell populations (**I**) are shown. Data represent or are pooled from two independent experiments (n=6 for AL T, AL NK, AL 7 R⁻ ILC1, 7R⁺ ILC1, and SP ILC1; n=3 for BM NK and BM iILC1). (**J**) The percentage of YFP⁺ cells in FL G1-ILCs in E18.5 straight PLZF-fm mice. Data are from one experiment (n=5). RNA-seq data are from two (AL NK cells and SG G1-ILCs), three (AL 7R⁺ ILC1s, BM iILC1s, and BM NK cells), and four (FL G1-ILCs, AL 7 R⁻ ILC1s, and LPL ILC1s) biological replicates (**E and F**). Data are presented as mean ± SD.

The online version of this article includes the following source data and figure supplement(s) for figure 1:

**Source data 1.** Fetal and adult liver contain bona fide ILC1s lacking IL-7R.

**Figure supplement 1.** Characterization of fetal and adult G1-ILC identities.

the development and maintenance of 7 R⁻ ILC1s are independent from 7R⁺ ILC1s in physiological conditions.

To test this hypothesis, we first explored whether there were molecular pathways controlling the development of each ILC1 population individually. RNA-seq revealed that 7R⁺ ILC1s highly expressed RORα and were positively enriched with gene sets 'RORA activates gene expression' relative to 7 R⁻ ILC1s, based on gene set enrichment analysis (GSEA) (*Figure 2D*). We therefore generated *Rora*⁻/⁻ mice to test the effect of RORα for 7R⁺ ILC1s. As *Rora*⁻/⁻ mice tend to die within 4 weeks after birth, we analyzed adult *Rora*⁺/⁻ mice or 2 weeks old *Rora*⁻/⁻ mice. 7R⁺ ILC1s were significantly reduced in *Rora*⁺/⁻ mice, while NK cells and 7 R⁻ ILC1s were unchanged (*Figure 2E*). In *Rora*⁻/⁻ mice, although whole ILC1s were significantly reduced in line with a recent study using *Ncr1*^Cre and *Vav1*^Cre *Rora*^fl/fl mice (*Song et al., 2021*), 7R⁺ ILC1s were the subset most apparently affected (*Figure 2F*). These data suggest that the development of 7 R⁻ ILC1s do not significantly depend on the presence of 7R⁺ ILC1s.

To make the developmental relationships between 7 R⁻ and 7R⁺ ILC1s clearer, we conducted adoptive transfer experiments under physiological conditions by using unirradiated CD45.1 WT host mice. 7 R⁻ and 7R⁺ ILC1s were isolated from AL, transferred, and the host liver were analyzed. For at least 2 months, little conversion was observed between 7 R⁻ and 7R⁺ ILC1s (*Figure 2G–I*), as evidenced by IL-7R and IL-18R1 expression (*Figure 2—figure supplement 1A*). In addition, transferred BM iILC1s gave rise to AL 7R⁺ ILC1s but not to 7 R⁻ ILC1s (*Figure 2—figure supplement 1B and C*), consistent with their transcriptional resemblance. However, parabiosis experiments showed that the replacement rate of AL 7R⁺ ILC1s were low (<5%), though significantly higher than that of 7 R⁻ ILC1s (*Figure 2—figure supplement 1D*), suggesting that both ILC1 subsets are tissue-resident. Thus, whether BM iILC1s actually contribute to AL 7R⁺ ILC1 pool is still unclear. To test the phenotypical stability of ILC1s in inflammatory states, we injected IL-15/IL-15Rα complex repeatedly into host mice that had received Cell Proliferation Dye (CPD) eFluor 450-labeled 7 R⁻ and 7R⁺ ILC1s. NK cells and 7R⁺ ILC1s proliferated more than 7 R⁻ ILC1s after the stimulation (*Figure 2J*), consistent with their basal Ki-67 expression levels (*Figure 1—figure supplement 1G*) and properties of cytokine responsiveness (*Figure 1—figure supplement 1E*). 7 R⁻ and 7R⁺ ILC1s were stable even after the IL-15/IL-15Rα stimulation (*Figure 2K*), confirming their stability in activated states. These results show that 7R⁺ ILC1s were rarely converted to 7 R⁻ ILC1s and not essential for the development of 7 R⁻ ILC1s under physiological conditions.

## FL G1-ILCs arise at hepatic parenchyma and give rise to 7R⁻ ILC1s in sinusoids

We next address the contribution of FL G1-ILCs to the development of AL 7 R⁻ ILC1s. Adoptively transferred FL G1-ILCs differentiated into CD49a⁺CD49b^lo mature ILC1s in AL (*Figure 3A*) and they completely lacked IL-7R (*Figure 3B*). To confirm the direct contribution of FL G1-ILCs to the adult pool of 7 R⁻ ILC1s, we performed fate-mapping experiments using Ncr1-CreERT2 Tg mice (*Nabekura and Lanier, 2016*) crossed with Rosa26-tdTomato mice. After tamoxifen injection into pregnant mice

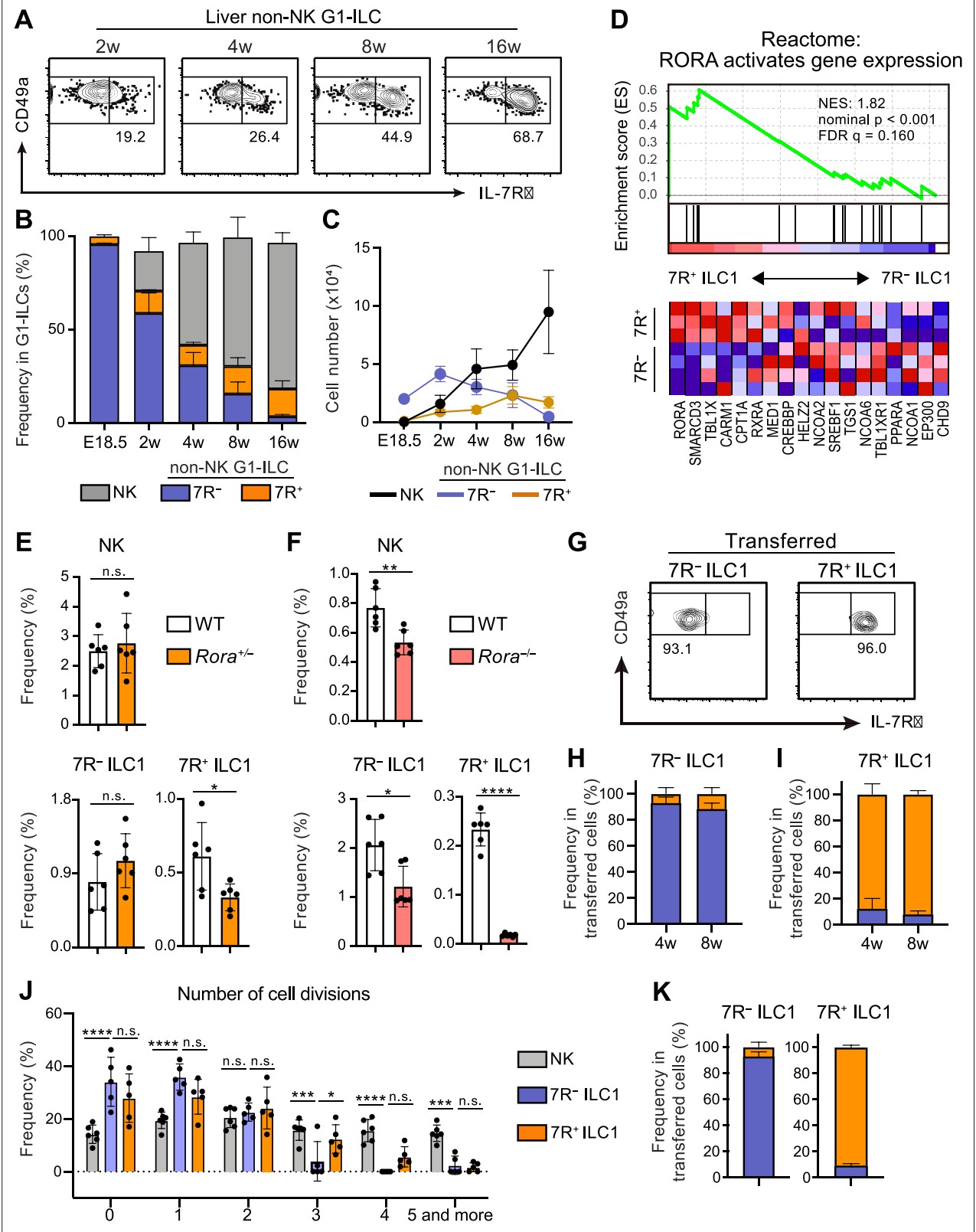

**Figure 2.** 7R+ ILC1 s are dispensable for the development of 7 R− ILC1s in AL. (A–C) Kinetics of IL-7Rα expression on liver non-NK G1-ILCs with age. Representative FCM profiles (**A**), the percentages within G1-ILCs (**B**), and the cell number (**C**) are shown. Data represent or are pooled from one (E18.5, n=6), two (2 w, n=5; 4 w, n=6; 8 w, n=9), and five (16 w, n=10) independent experiments. (**D**) GSEA of transcriptomes in 7R+ ILC1s compared to 7 R− ILC1s. Eighteen genes included in an indicated gene set from Reactome Pathway are shown. The lower heatmap shows relative gene expression levels

*Figure 2 continued on next page*

*Figure 2 continued*

in AL 7 R⁻ and 7R⁺ ILC1s. (**E**) The percentages of AL G1-ILC populations in control or *Rora*⁺/⁻ mice. Data are pooled from two independent experiments (WT, n=6; *Rora*⁺/⁻, n=6). (**F**) The percentages of AL G1-ILC populations in control or *Rora*⁻/⁻ mice. Data are from one experiment (WT, n=6; *Rora*⁻/⁻, n=6). (**G–I**) Flow cytometric (FCM) analysis of transferred AL 7 R⁻ and 7R⁺ ILC1s detected in the host liver at 4 weeks and 8 weeks post-transfer. Representative FCM profiles at 4 weeks post-transfer (**G**) and the percentages of the fate of transferred 7 R⁻ ILC1s (**H**) and 7R⁺ ILC1s (**I**) are shown. Data represent or are pooled from two (7 R⁻ ILC1 8 w, 7R⁺ ILC1 8w; n=3) and three (7 R⁻ ILC1 4 w, n=6; 7R⁺ ILC1 4w; n=4) independent experiments. (**J and K**) Host mice received with CPD-eFluor 450-labeled 7 R⁻ and 7R⁺ ILC1s are stimulated with i.p. injection of IL-15/IL-15Rα complex at days 1, 3, and 5. After 7 days, liver leukocytes of recipient mice were analyzed. Frequency of how many times each cell divides are calculated based on the FCM analysis of CPD-eFluor 450 dye dilution (**J**) (two independent experiments; n=6 for NK, n=5 for 7 R⁻ ILC1 and 7R⁺ ILC1) and the percentages of the fate of transferred AL 7 R⁻ ILC1s and 7R⁺ ILC1s (**K**) (two independent experiments; n=6 for 7 R⁻ ILC1, n=5 for 7R⁺ ILC1) are shown. RNA-seq data are from three (AL 7R⁺ ILC1s) and four (AL 7 R⁻ ILC1s) biological replicates (**D**). Data are presented as mean ± SD. *p<0.05, **p<0.01, ***p<0.001, ****p<0.0001.

The online version of this article includes the following source data and figure supplement(s) for figure 2:

**Source data 1.** 7R⁺ ILC1s are dispensable for the development of 7 R⁻ ILC1s in AL.

**Figure supplement 1.** BM iILC1s have ability to give rise to AL 7*R*⁺ ILC1 s in vivo.

at E17.5, liver of neonatal and 4 weeks old pups were analyzed. TdTomato expression was clearly restricted to IL-7R⁻ fractions within neonatal G1-ILCs and AL ILC1s (*Figure 3C–E*, *Figure 3—figure supplement 1A*), consistent with previous studies (*Chen et al., 2022*; *Sparano et al., 2022*). As shown in these studies, fate-mapped 7 R⁻ ILC1s showed a skewed expression of Ly49E/F, though they also contained Ly49E/F⁻ population (20–25%) (*Figure 3—figure supplement 1B–1D*). Labeling efficiency was 40% in neonatal IL-7R⁻ G1-ILCs and 20% in 7 R⁻ ILC1s in 4 weeks old mice (*Figure 3F and G*). These results confirm a direct, albeit partial, contribution of FL G1-ILCs to the adult pool of 7 R⁻ ILC1s.

To investigate the detailed developmental process of FL G1-ILCs and AL ILC1s in vivo, we examined their spatiotemporal distributions. In immunostaining analysis, FL G1-ILCs were identified as NKp46⁺ cells in WT mice (*Figure 4A*). AL NK cells and ILC1s were identified as NKp46⁺GFP⁻ and NKp46⁺GFP⁺ cells in *Cxcr6*^GFP/+ mice, respectively (*Figure 4B* and *Figure 5—figure supplement 1A*). In E18.5 liver, FL G1-ILCs mostly distributed at perivascular sites, outside of the sinusoidal lumen (here termed parenchyma; *Figure 4A*). In contrast, over 85% of whole ILC1s and NK cells were within sinusoids in AL (*Figure 4B and C*). Although we could not detect IL-7R expression on ILC1s by immunofluorescence, flow cytometry (FCM)-based analysis of intravenous (i.v.) CD45.2 staining confirmed similar intravascular locations of 7 R⁻ and 7R⁺ ILC1s as well as NK cells, T cells, and NKT cells in AL (*Figure 4D and E*). In contrast, ILC2s were not efficiently labeled by i.v. staining, consistent with perivascular localization of liver ILC2s observed so far (*Dahlgren et al., 2019*). Interestingly, a population of AL Lin⁻Sca-1⁺Mac-1⁺ (LSM) cells, local precursors for ILC1s (*Bai et al., 2021*), were also not well labeled by i.v. staining. Thus, there are localization shifts between ILC1 precursors and mature ILC1s in the liver: FL G1-ILCs and some LSM cells distribute to parenchyma, whereas AL G1-ILCs including 7 R⁻ ILC1s reside within sinusoids. These observations suggest that FL G1-ILCs arise at parenchyma and then infiltrate into sinusoids during maturation toward 7 R⁻ ILC1s.

## Hepatocyte-derived IL-15 supports the parenchymal development of 7R⁻ ILC1s

Given that the possible origins of FL G1-ILCs and 7 R⁻ ILC1s are liver parenchyma, we asked the role of parenchymal microenvironments for ILC1 development. Since IL-15, transpresented as an IL-15/IL-15Rα complex, is a local determinant of G1-ILC homeostasis (*Ikuta et al., 2021*), we generated *Il15*^fl/fl mice and crossed them with several Cre-driver lines. Reanalysis of single nuclei RNA-seq (snRNA-seq) data of whole liver cells from Liver Cell Atlas (https://www.livercellatlas.org/) revealed that *Il15* gene was highly expressed by macrophages and endothelial cells and, to a lesser extent, by hepatocyte, while *Il15ra* expression was prominent in hepatocytes (*Figure 5A and B*, and *Figure 5—figure supplement 1B*). We therefore focused on IL-15 produced by hepatocytes, macrophages, and endothelial cells. To target parenchymal IL-15, we generated Alb-Cre *Il15*^fl/fl mice, which lacked IL-15 in hepatocytes. In Alb-Cre *Il15*^fl/fl mice, FL G1-ILCs were significantly reduced (*Figure 5C*). Notably, Alb-Cre *Il15*^fl/fl mice also showed reduction of AL 7 R⁻ ILC1s in contrast to unchanged NK cells, 7R⁺ ILC1s, and NKT cells (*Figure 5D and E*, and *Figure 5—figure supplement 1C*), despite their similar intravascular localizations. These data corroborate the precursor-progeny relationship of FL G1-ILCs and AL 7 R⁻ ILC1s and also the parenchymal origin of 7 R⁻ ILC1s. To further define the IL-15 niches

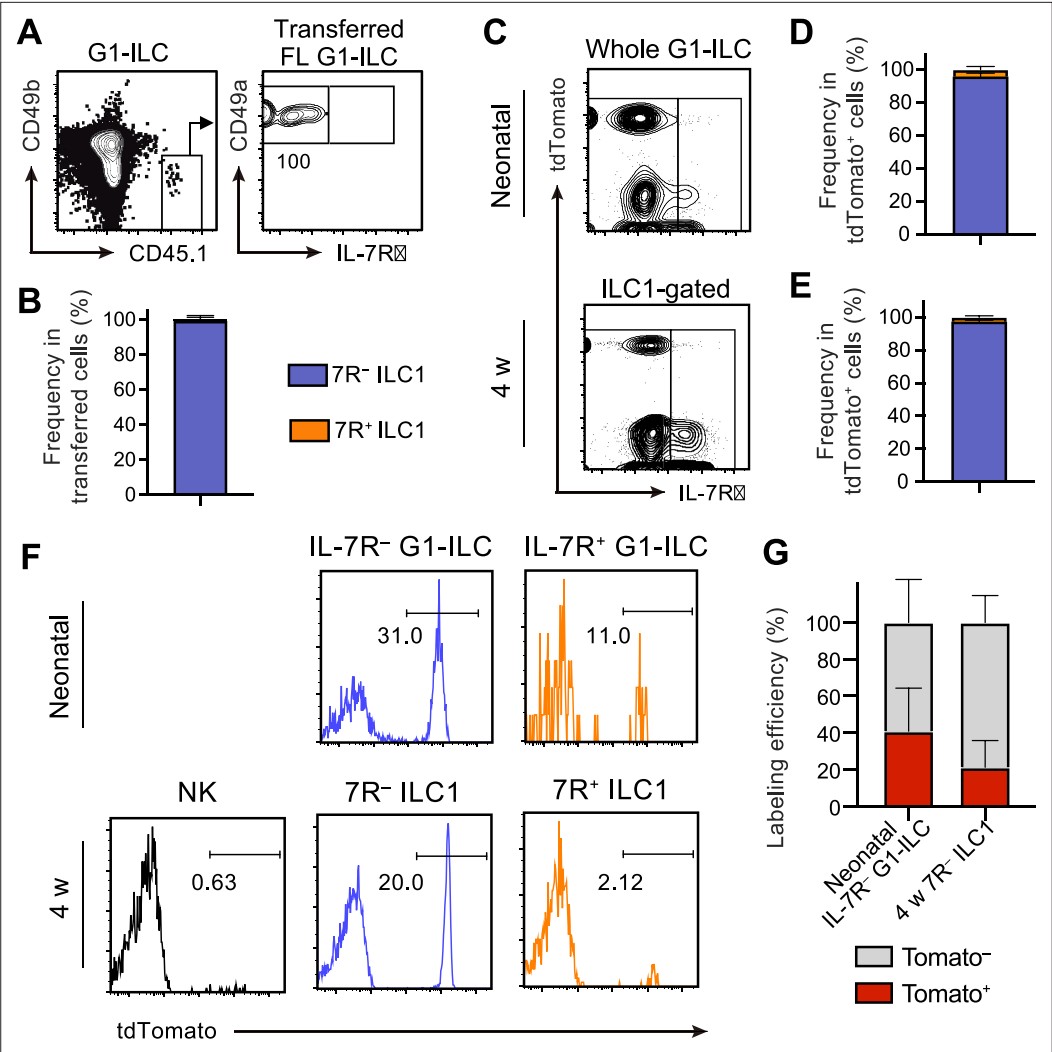

**Figure 3.** FL G1-ILCs exclusively give rise to 7 R⁻ ILC1s. (**A and B**) FCM analysis of transferred FL G1-ILCs (CD45.1) detected in the host liver (CD45.2) at 4 weeks post-transfer. Representative FCM profiles (**A**) and the percentages of transferred cell fate (**B**) are shown. Data represent or are pooled from two independent experiments (n=4). (**C–E**) FCM analysis of tdTomato⁺ cells in neonatal and adult Ncr1-CreERT2 Rosa26-tdTomato mice treated with tamoxifen at E17.5. Representative FCM plots (**C**) and the percentage of the fate of tdTomato⁺ cells in neonates (**D**) and 4 weeks old mice (**E**) are shown. Blue, IL-7R⁻ fraction; orange, IL-7R⁺ fraction. Data represent or are pooled from three independent experiments (neonatal, n=10; 4 w, n=5). (**F and G**) FCM analysis of tdTomato expression on indicated G1-ILC populations in Ncr1-CreERT2 Rosa26-tdTomato mice treated with tamoxifen at E17.5. Representative histograms (**F**) and the percentages of tdTomato⁺ and tdTomato⁻ fractions in neonatal IL-7R⁻ G1-ILCs and 7 R⁻ ILC1s in 4 weeks old mice (**G**) are shown. Data represent or are pooled from three independent experiments (neonatal, n=10; 4 w, n=5). Data are presented as mean ± SD.

The online version of this article includes the following source data and figure supplement(s) for figure 3:

**Source data 1.** FL G1-ILCs exclusively give rise to 7 R⁻ ILC1s.

**Figure supplement 1.** FL G1-ILCs contribute to AL 7 R⁻ ILC1 pool.

for G1-ILCs, we generated *Lyve1*^Cre/+ *Il15*^fl/fl mice, which target vascular IL-15 sources including sinusoidal endothelial cells and a fraction of hematopoietic cells (*Lim et al., 2018*; *Pham et al., 2010*). In *Lyve1*^Cre/+ *Il15*^fl/fl mice, all AL G1-ILC subsets were significantly reduced (*Figure 5F*). We analyzed another mouse line targeting intravascular IL-15 sources, Lyz2-Cre *Il15*^fl/fl mice, which lack IL-15 in myeloid cells. Lyz2-Cre *Il15*^fl/fl mice showed similar two-fold reductions of AL NK cells, 7 R⁻ ILC1s, and 7R⁺ ILC1s (*Figure 5G*), confirming the similar IL-15 requirements among all G1-ILC subsets. Notably, expression of Bcl-2, a survival factor downstream of IL-15, was downregulated in all G1-ILCs

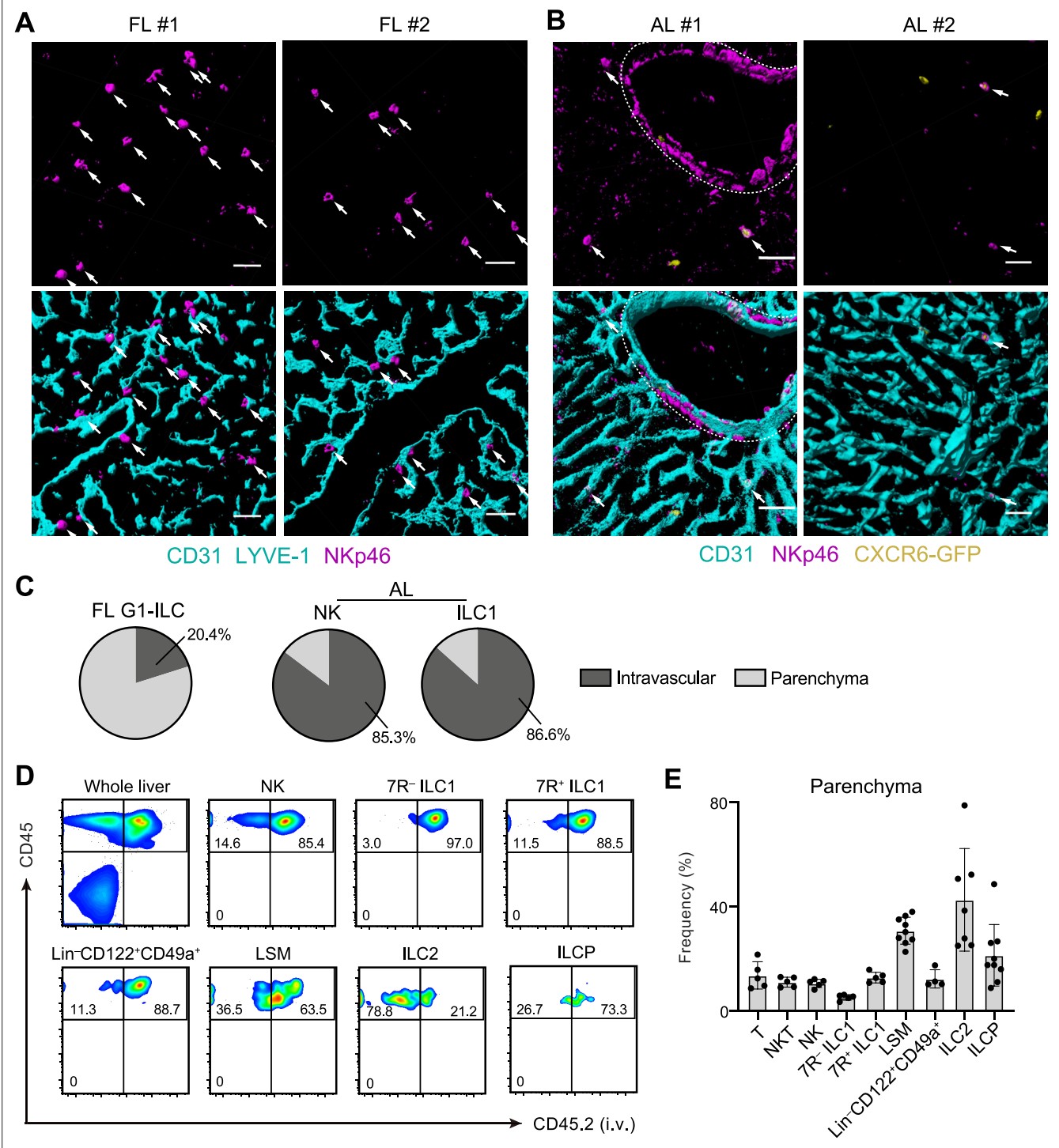

**Figure 4.** Liver G1-ILCs shift distributions from parenchyma to sinusoids during development. (**A and B**) 3D-reconstructed immunofluorescence images of frozen sections of FL from WT mice (A; n=4) and AL from *Cxcr6*GFP/+ mice (B; n=4) stained with anti-NKp46 (magenta) and anti-CD31 and/or anti-LYVE-1 (for FL endothelium) (cyan) antibodies. GFP signals are shown in yellow. White arrows indicate G1-ILCs. Hepatic artery is circled by a dotted line. Scale bar, 40 μm. (**C**) The percentages of indicated G1-ILC subsets localized inside (intravascular) or outside (parenchyma) of the blood vessels. Data represent randomly counted 323 cells for FL G1-ILCs pooled from four E18.5 WT mice and 232 cells for AL NK cells as well as 136 cells for AL ILC1s pooled from four *Cxcr6*GFP/+ mice. (**D and E**) FCM analysis of AL of mice injected i.v. with PE/Cy7 anti-CD45.2 antibody 2 min before the liver perfusion and leukocyte isolation. Representative FCM profiles (**D**) and the percentages of cells unlabeled by i.v. CD45.2 staining within CD45+ cells (considered as parenchyma-distributed cells) (**E**) are shown. Data represent or are pooled from two (n=5 for T, NKT, NK, 7R+ ILC1, and 7 R− ILC1), four (n=7 for ILC2 and Lin−CD49a+CD122+ cells), and five (n=9 for ILCP and LSM cells) independent experiments. Data are presented as mean ± SD.

*Figure 4 continued on next page*

*Figure 4 continued*

The online version of this article includes the following source data for figure 4:

**Source data 1.** Liver G1-ILCs shift distributions from parenchyma to sinusoids during development.

of *Lyve1*$^{Cre/+}$ *Il15*$^{fl/fl}$ mice (*Figure 5H and I*), whereas Bcl-2 and Ki-67 levels were unchanged in Alb-Cre *Il15*$^{fl/fl}$ mice (*Figure 5J and K*). These results indicate that parenchymal IL-15 has no direct impact on mature 7 R⁻ ILC1s in sinusoids whereas intravascular IL-15 directly supports the survival of all AL G1-ILCs. Alb-Cre *Il15*$^{fl/fl}$ mice had reduced Lin⁻CD122⁺CD49a⁺ ILC1 precursors in AL (*Figure 5—figure supplement 1D*), suggesting an impaired development of 7 R⁻ ILC1s. Thus, these data demonstrate that hepatocyte-derived IL-15 supports the development of 7 R⁻ ILC1s at parenchyma, thereby maintaining AL 7 R⁻ ILC1s infiltrated in sinusoids.

## Steady-state mTOR activity confers granzyme B-mediated cytotoxicity in 7R⁻ ILC1s

Cytotoxicity is one of the most pivotal functions of G1-ILCs, though the contribution of ILC1s remains controversial. By focusing on the ILC1 heterogeneity, we attempted to describe the G1-ILC effector function in detail. In steady state, minimal levels of granzyme B and death ligands were found on NK cells, while 7 R⁻ ILC1s expressed both granzyme B and TRAIL, the latter of which was also expressed on 7R⁺ ILC1s (*Figure 6A and B*, *Figure 6—figure supplement 1A, B*; *Friedrich et al., 2021*). In line with this, freshly isolated 7 R⁻ ILC1s remarkably lysed multiple tumor cells including YAC-1 (*Figure 6C*), Hepa1-6 (*Figure 6D*), and B16F10 cells (*Figure 6E*). By contrast, NK cells and 7R⁺ ILC1s showed only slight or no cytotoxicity against these tumor cells, consistent with a previous study showing minimal cytotoxicity of unstimulated NK cells (*Fehniger et al., 2007*). To determine the effector pathways 7 R⁻ ILC1s rely on, we added concanamycin A (CMA), an inhibitor for perforin/granzyme pathways (*Kataoka et al., 1994*), and neutralizing antibodies for TRAIL and FasL to the coculture systems. Killing of Hepa1-6 cells by 7 R⁻ ILC1s was markedly inhibited by CMA, and to a lesser extent by anti-TRAIL antibody (*Figure 6F*). Despite the expression of granzyme A in NK cells and granzyme C in 7R⁺ ILC1s (*Nixon et al., 2022*; *Figure 6—figure supplement 1C and D*), CMA had no effect to their cytotoxicity. Other granzyme genes (*Gzmf, k, n*, and *m*) were undetectable in ILC1s (data not shown). These results suggest that granzyme B plays a major role in the cytotoxicity of 7 R⁻ ILC1s.

Cellular amount of granzyme B is well correlated to and primarily responsible for the NK cell cytotoxicity (*Bhat et al., 2007*; *Gwalani and Orange, 2018*; *Prager et al., 2019*). Although granzyme B expression and killing capacity of NK cells are weak at steady state, stimulation by cytokines, especially by IL-15, enable to induce both of them (*Fehniger et al., 2007*; *Marçais et al., 2014*; *Prager et al., 2019*). To test whether 7 R⁻ ILC1s share such an activation machinery, we analyzed their expression of effector molecules after the stimulation. Expression levels of granzyme B, as well as TRAIL and granzyme C, in each G1-ILC subset were clearly upregulated by in vitro stimulation of IL-15 (*Figure 6—figure supplement 1E*) and in vivo injection of IL-15/IL-15Rα complex (*Figure 6A and B*, *Figure 6—figure supplement 1A*, and 1B). Interestingly, however, both IL-15 stimulation and IL-15/IL-15Rα injection enhanced granzyme B more efficiently on NK cells than 7 R⁻ ILC1s, and thereby the granzyme B levels of NK cells overwhelmed or got comparable to those of 7 R⁻ ILC1s (*Figure 6A* and *Figure 6—figure supplement 1E*). IL-15/IL-15Rα injection also triggered the phosphorylation of STAT5, Akt, and ribosomal protein S6 (a target of mTOR), which are critical for the IL-15-induced effector function (*Ali et al., 2015*), in NK cells and 7R⁺ ILC1s but to a lesser degree in 7 R⁻ ILC1s (*Figure 6G and H*, *Figure 6—figure supplement 1F*, and 1 G). These data suggest that the cytotoxic capacity of 7 R⁻ ILC1s are different from that of NK cells in terms of responsiveness and requirement for the cytokine stimulation. Interestingly, the phosphorylation level of S6 was rather higher in ILC1s than NK cells in unstimulated mice (*Figure 6G and H*). Notably, injection of rapamycin, an mTOR complex inhibitor, downregulated granzyme B expression in 7 R⁻ ILC1s to a level comparable to that in NK cells (*Figure 6I and J*). By contrast, granzyme B levels in NK cells and 7R⁺ ILC1s were unaffected. Collectively, these results show that 7 R⁻ ILC1s exhibit cytotoxicity in their steady state through granzyme B expression, which is supported by their constitutive mTOR activation.

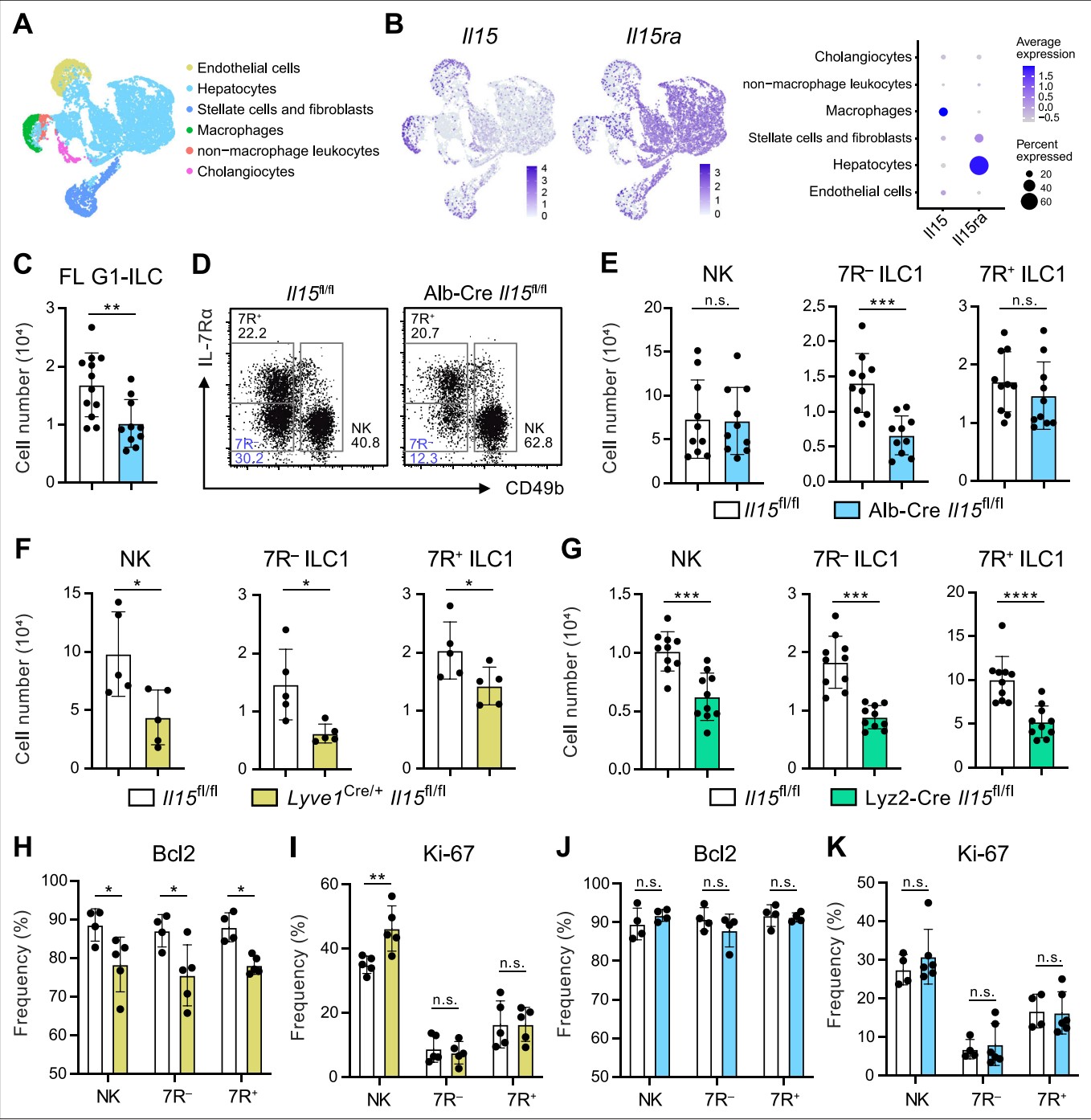

**Figure 5.** Hepatocytes provide the parenchymal IL-15 niche regulating the local development of 7 R⁻ ILC1s. (**A and B**) Single nuclei RNA-seq (snRNA-seq) analysis of mouse whole liver cells (Liver Cell Atlas; https://www.livercellatlas.org/). UMAP visualization (**A**) and expression levels of *Il15* and *Il15ra* (**B**) in each cell population assigned in *Figure 5—figure supplement 1B* are shown. (**C**) The cell number of FL G1-ILCs in control or Alb-Cre *Il15*fl/fl mice. Data are pooled from two independent experiments (*Il15*fl/fl, n=12; Alb-Cre *Il15*fl/fl, n=10). (**D and E**) FCM analysis of AL G1-ILCs in control or Alb-Cre *Il15*fl/fl mice. Representative FCM plots (**D**) and the cell number of each population (**E**) are shown. Data represent or are pooled from two independent experiments (*Il15*fl/fl, n=10; Alb-Cre *Il15*fl/fl, n=10). (**F and G**) The cell number of indicated G1-ILC populations in *Lyve1*Cre/+ *Il15*fl/fl mice (**F**) (two independent experiments; n=5 for *Il15*fl/fl and *Lyve1*Cre/+ *Il15*fl/fl mice) or Lyz2-Cre *Il15*fl/fl mice (**G**) (two independent experiments; n=10 for *Il15*fl/fl and Lyz2-Cre *Il15*fl/fl mice) compared to controls. (**H and I**) The percentages of Bcl-2 (**H**) (two independent experiments; n=4 for *Il15*fl/fl, n=5 for *Lyve1*Cre/+ *Il15*fl/fl mice) and Ki-67 (**I**) (two independent experiments; n=5 for *Il15*fl/fl, n=5 for *Lyve1*Cre/+ *Il15*fl/fl mice) expressing cells within each G1-ILC population in control or *Lyve1*Cre/+ *Il15*fl/fl mice. (**J and K**) The percentages of Bcl-2 (**J**) (two independent experiments; n=4 for *Il15*fl/fl, n=4 for Alb-Cre *Il15*fl/fl mice) and Ki-67 (**K**) (two

*Figure 5 continued on next page*

*Figure 5 continued*

independent experiments; n=4 for *Il15*^fl/fl, n=6 for Alb-Cre *Il15*^fl/fl mice) expressing cells within each G1-ILC population in control or Alb-Cre *Il15*^fl/fl mice. Data are presented as mean ± SD. *p<0.05, **p<0.01, ***p<0.001, ****p<0.0001.

The online version of this article includes the following source data and figure supplement(s) for figure 5:

**Source data 1.** Hepatocytes provide the parenchymal IL-15 niche regulating the local development of 7 R⁻ ILC1s.

**Figure supplement 1.** Liver IL-15-producing cells supports lymphoid cells in a subset-dependent manner.

## Discussion

In this study, we have characterized the developmental process and functional heterogeneity of liver G1-ILCs. Hepatocytes shape IL-15 niches supporting parenchymal development of FL G1-ILCs, that differentiate into 7 R⁻ ILC1s in sinusoids. Functionally, 7 R⁻ ILC1s exhibit granzyme B-mediated cytotoxicity in steady state, in sharp contrast to less cytotoxic resting NK cells.

ILC1 heterogeneity has been extensively addressed recently. In the liver, ILC1s are separated into IL-7R⁻ and IL-7R⁺ populations (*Friedrich et al., 2021*; *Sparano et al., 2022*; *Yomogida et al., 2021*), the latter of which can differentiate into the former when cultured in vitro or transferred into lymphopenic mice (*Friedrich et al., 2021*). However, we show that 7 R⁻ and 7R⁺ ILC1s behave like independent subsets under physiological conditions: decline of 7 R⁻ ILC1s and accumulation of 7R⁺ ILC1s with age, requirements for RORα specifically in 7R⁺ ILC1s, and phenotypical stability between 7 R⁻ and 7R⁺ ILC1s when transferred into WT host mice. Such a contradiction might be due to the highly nutrient- and cytokine-accessible environments in the culture systems and lymphopenic hosts that might trigger non-physiological activation and phenotypic shift of 7R⁺ ILC1s. Indeed, our model rather gives an explanation for the ILC1 heterogeneity in inflammatory disease models using healthy mice observed so far. In mouse models of contact hypersensitivity, MCMV infection, and liver injury, ILC1s with high expression of cytokine receptors (IL-7R, CD25, and/or IL-18R) highly proliferate and accumulate in AL, thereby forming the memory and protecting liver from infection and injury (*Nabekura et al., 2020*; *Wang et al., 2018*; *Weizman et al., 2019*). AL 7R⁺ ILC1s resemble such 'memory-like' or 'activated' ILC1s in terms of the surface phenotype and high proliferation potentials. These observations suggest a hypothesis that a preferential proliferation of pre-existing stable 7R⁺ ILC1s, rather than inflammation-specific ILC1s induced from naïve ILC1s, may contribute to liver immunity and homeostasis.

Several previous studies have pointed out the precursors for ILC1s: BM iILC1s (*Klose et al., 2014*), FL G1-ILCs (*Constantinides et al., 2015*; *Daussy et al., 2014*), and local precursors in the liver such as LSM cells and Lin⁻CD122⁺CD49a⁺ cells (*Bai et al., 2021*). In particular, FL G1-ILCs are precursors for AL Ly-49E⁺ ILC1s (*Chen et al., 2022*; *Sparano et al., 2022*). Although AL Ly-49E⁺ ILC1s are included in and account for 30–40% of AL 7 R⁻ ILC1s, fate-mapping reveals that FL-derived 7 R⁻ ILC1s contain also an Ly49E/F⁻ population (20–25%), suggesting further heterogeneity in FL-derived ILC1s. Considering the partial contribution (about 50%) of FL G1-ILCs to AL 7 R⁻ ILC1 pool estimated by fate-mapping, local ILC1 precursors such as LSM cells and Lin⁻CD122⁺CD49a⁺ cells might be the other sources for 7 R⁻ ILC1s. By contrast, the origin of AL 7R⁺ ILC1s remains to be solved. We show that BM iILC1s have a potential to differentiate into AL 7R⁺ ILC1s, but the actual contribution is unclear. As both AL 7 R⁻ and 7R⁺ ILC1s were rarely replaced during parabiosis experiments using adult mice, it is possible that a transiently migrated population derived from BM settle and give rise to 7R⁺ ILC1s in the liver during neonatal period, as discussed previously (*Sparano et al., 2022*). Further investigations using a specific tracing approach such as fate-mapping of BM iILC1s are required to determine their precise developmental potency.

Development and maintenance of ILCs strictly depend on their resident tissue microenvironments, called niche (*Ikuta et al., 2021*; *Kotas and Locksley, 2018*; *McFarland and Colonna, 2020*; *Murphy et al., 2022*). IL-15 is a cytokine crucial for G1-ILC homeostasis. IL-15-producing cells shape the niches for G1-ILCs within various tissues via the IL-15/IL-15Rα transpresentation (*Cepero-Donates et al., 2016*; *Cui et al., 2014*; *Liou et al., 2014*; *Mortier et al., 2009*), yet IL-15 niches specific for ILC1s remained unclear. Using a combination of imaging and cell-specific IL-15 knockout approaches, we unveil the parenchymal origins of 7 R⁻ ILC1s as well as FL G1-ILCs and identify hepatocytes as an IL-15-producing niche supporting 7 R⁻ ILC1 development. Parenchymal distribution of FL G1-ILCs, reminiscent of FL hematopoietic stem cells (HSCs) (*Khan et al., 2016*; *Lewis et al., 2021*), might be

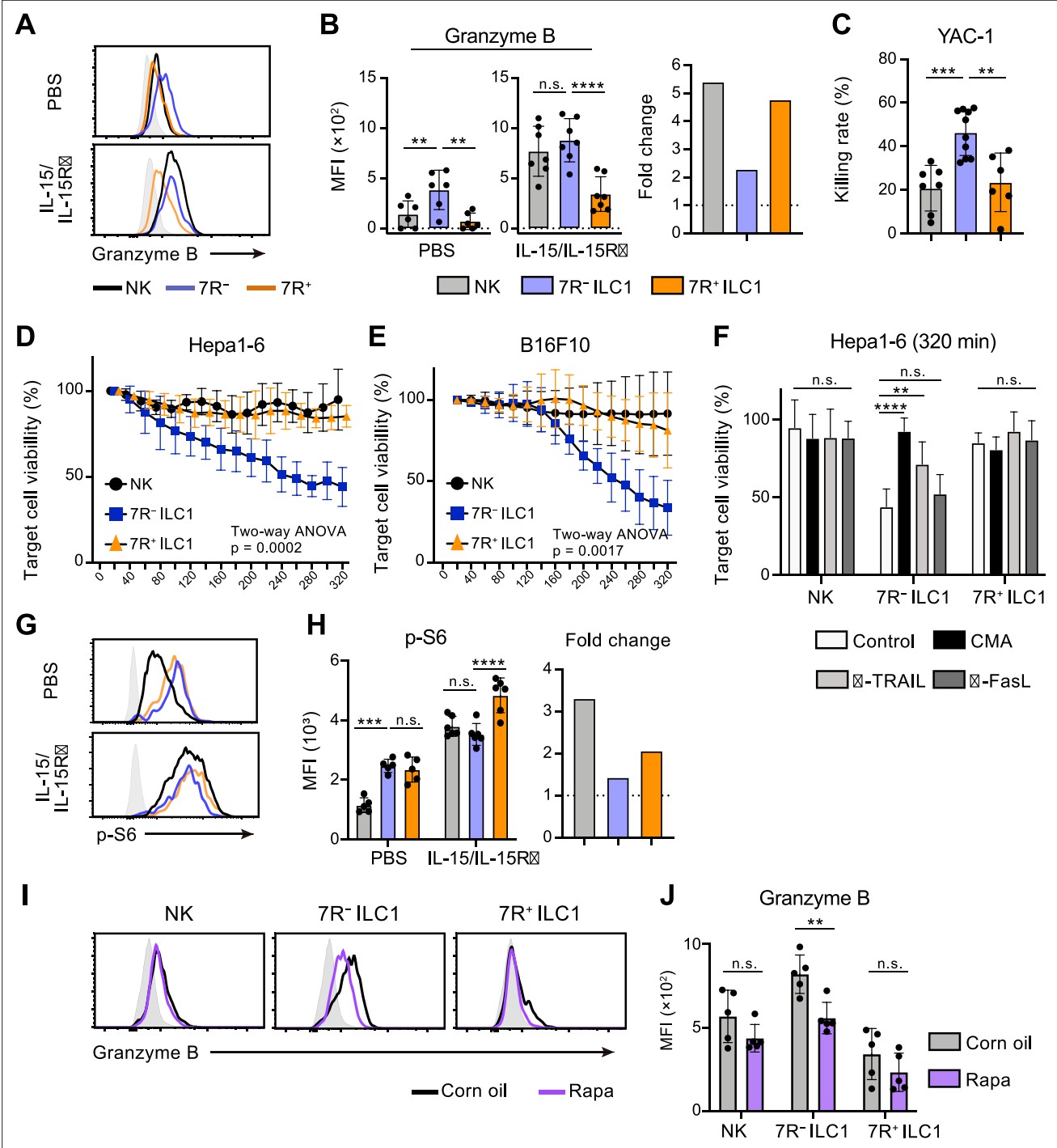

**Figure 6.** 7 R⁻ ILC1s exhibit cytotoxicity via granzyme B expression underpinned by steady-state mTOR activation. (**A and B**) Granzyme B expression on each AL G1-ILC population in control or IL-15/IL-15Rα-treated mice. Representative histograms (**A**), MFI (B, left), and its fold change after the IL-15/IL-15Rα treatments (B, right) are shown. Shaded histograms (grey) indicate isotype controls. Data represent or are pooled from three independent experiments (PBS, n=6; IL-15/IL-15Rα, n=7). (**C**) The percentages of annexin V⁺PI⁺ YAC-1 cells in flow-based cytotoxicity assays. Freshly isolated effector cells were co-cultured with target cells for 4 hours (E:T ratio = 10:1). Data are pooled from two independent experiments (NK, n=7; 7 R⁻ ILC1, n=10; 7R⁺ ILC1, n=6). (**D and E**) Target cell viability at each timepoint in time-lapse cytotoxicity assay using Hepa1-6 cells (**D**) (two independent experiments; n=6 for NK, n=6 for 7 R⁻ ILC1, and n=4 for 7R⁺ ILC1 in all timepoints) and B16F10 cells (**E**) (two independent experiments; n=7 for NK, n=8 for 7 R⁻ ILC1, and n=8 for 7R⁺ ILC1 in all timepoints) as target cells. Freshly isolated effector cells were co-cultured with target cells up to 6 hours (E:T ratio = 10:1). (**F**) Hepa1-6 cell viability at 320 min in time-lapse cytotoxicity assays supplemented with concanamycin A (CMA) or neutralizing antibody for TRAIL (α-TRAIL) or FasL (α-FasL) compared to vehicle-supplemented controls. Data are pooled from two independent experiments (NK, n=6; 7 R⁻ ILC1, n=6;

*Figure 6 continued on next page*

*Figure 6 continued*

7R$^+$ ILC1, n=4). (**G and H**) FCM analysis of phosphorylation of S6 in NK cells (black), 7 R$^-$ ILC1s (blue), and 7R$^+$ ILC1s (orange) in control or IL-15/IL-15Rα-treated mice. Representative histograms (**G**), MFI (left), and its fold change after the IL-15/IL-15Rα treatments (right) (**H**) are shown. Shaded histograms (grey) indicate isotype controls. Data represent or are pooled from three independent experiments (PBS, n=5; IL-15/IL-15Rα, n=6). (**I and J**) FCM analysis of granzyme B expressed on each AL G1-ILC population in control or rapamycin-treated (rapa) mice. Representative histograms (**I**) and MFI (**J**) are shown. Shaded histograms (grey) indicate isotype controls. Data represent or are pooled from three independent experiments (corn oil, n=5; rapa, n=5). Data are presented as mean ± SD. **p<0.01, ***p<0.001, ****p<0.0001.

The online version of this article includes the following source data and figure supplement(s) for figure 6:

**Source data 1.** 7 R$^-$ ILC1s exhibit cytotoxicity via granzyme B expression underpinned by steady-state mTOR activation.

**Figure supplement 1.** Differential effector molecule expression and cytokine responsiveness among heterogenous G1-ILC subsets.

partly due to the immaturity of hepatic vasculature in that period. Since FL G1-ILCs and FL HSCs are also similar in that they eventually infiltrate into blood vessels (*Lewis et al., 2021*), it would be of interest to address the mechanism underlying their neonatal dynamics. In addition, it is still unclear why hepatocyte-derived IL-15 has such a local effect despite many fenestrae and the lack of a basement membrane on liver sinusoids. One possible explanation is that the transpresentation of IL-15/IL-15Rα by hepatocytes may require direct contact to target cells, as dendritic cells do (*Mortier et al., 2008*). Given that hepatocytes prominently express *Il15ra* gene and its deletion results in the reduction of whole IL-15-dependent lymphocytes in the liver (*Cepero-Donates et al., 2016*), it is also possible that hepatocytes may produce IL-15Rα as a soluble form, that binds to other cell-derived IL-15 to exert non-local effects.

Traditionally, ILC1s are regarded as less cytotoxic than NK cells in mice, whereas recent studies have challenged this theory (*Dadi et al., 2016*; *Di Censo et al., 2021*; *Kansler et al., 2022*; *Nixon et al., 2022*; *Yomogida et al., 2021*). Our study provides two possible explanations for this discrepancy. First, the age of mice selected for analysis influence the composition and overall cytotoxicity of ILC1s. We and others (*Chen et al., 2022*; *Friedrich et al., 2021*; *Nixon et al., 2022*) showed that ILC1s were heterogenous in their cytotoxicity. Due to the age-dependent reduction of highly cytotoxic 7 R$^-$ ILC1s, the overall cytotoxicity of ILC1s in the liver should decline with age. Second, the effector program of 7 R$^-$ ILC1s differ from NK cells in its nature, especially in terms of cytokine responsiveness. Freshly isolated NK cells exhibit low expression of cytotoxic molecules and only minimal cytotoxicity (*Fehniger et al., 2007*), while stimulation by IL-15 confers granzyme B expression and cytotoxicity on NK cells via mTOR-dependent metabolic reprogramming (*Marçais et al., 2014*; *Nandagopal et al., 2014*). Conversely, we show that 7 R$^-$ ILC1s exhibit prominent granzyme B-mediated cytotoxicity via mTOR activity at steady state, though they are less responsive to cytokines than NK cells and 7R$^+$ ILC1s. These findings suggest that 7 R$^-$ ILC1s are 'ready-to-kill' sentinels that contribute to the tonic immune surveillance, which is followed later by the response of activated and proliferated NK cells and 7R$^+$ ILC1s. While the meaning of the predominance of such a cytotoxic subset especially in early life is unknown, it is possible that FL G1-ILCs and 7 R$^-$ ILC1s play some physiological roles in the early liver development or maturation by eliminating unnecessary cells.

Taken together, our study provides insight into the complex ILC1 ontogeny by revealing relationships among heterogenous ILC1 subsets, their developmental dynamics, and niche dependence. Our findings highlight the intrinsic cytotoxic programs of 7 R$^-$ ILC1s unlike NK cells, proposing them as critical steady-state sentinels against infection prevention and tumor surveillance and bringing the possibility of local therapeutic targeting of ILC1 function.

# Materials and methods
## Mice

C57BL/6 J mice were purchased from Japan SLC (Hamamatsu, Japan). *Il7*$^{-/-}$ mice were obtained by *Il7*$^{fl/fl}$ mice developed in our laboratory (*Liang et al., 2012*) with Cre-mediated germ-line deletion. *Zbtb16*$^{IRES-EGFP-Cre}$ knock-in (KI) mice (PLZF$^{GFPcre}$ mice) (*Constantinides et al., 2014*) were provided by Dr. M. Miyazaki at Kyoto University and crossed with Rosa26-YFP mice (*Srinivas et al., 2001*). RORα knockout (*Rora*$^{-/-}$) mice were generated by CRISPR/Cas9 gene editing in our laboratory and will be reported in detail elsewhere. *Cxcr6*$^{GFP/+}$ KI mice were provided by Dr. H. Ohno. Alb-Cre transgenic (Tg) mice (*Postic et al., 1999*) kindly supplied by Dr. Mark A. Magnuson at Vanderbilt University, *Lyve1*$^{Cre}$ KI

mice (*Pham et al., 2010*) kindly supplied by Dr. Jason Cyster at University of California San Francisco, and Lyz2-Cre Tg mice were bred with *Il15*<sup>flox/flox</sup> (*Il15*<sup>fl/fl</sup>) mice, which were generated in our laboratory (Cui et al., under review). Ncr1-CreERT2 Tg mice (*Nabekura and Lanier, 2016*) were provided by Dr. T. Nabekura and Dr. Lewis L. Lanier and crossed with Rosa26-tdTomato (*Madisen et al., 2010*) mice. For fetal experiments, the noon when the vaginal plug was detected was considered as embryonic day (E) 0.5. All mice were maintained under specific pathogen-free conditions in the Experimental Research Center for Infectious Diseases at the Institute for Life and Medical Sciences, Kyoto University. All procedures were carried out under sevoflurane or isoflurane anesthesia to minimize animal suffering. All mouse protocols were approved by the Animal Experimentation Committee of the Institute for Life and Medical Sciences, Kyoto University.

## Cell preparation and isolation

To protect ILC1s from NAD⁺-induced cell death (NICD) (*Stark et al., 2018*), mice were intravenously (i.v.) injected with 40 µg ARTC2.2 blocking nanobody (BioLegend, San Diego, CA, USA) 30 min before sacrificing the mice in several experiments. Fetal liver, adult liver, spleen, peripheral (axillary, brachial, and inguinal) lymph nodes, and mesenteric lymph nodes were dissociated mechanically and passed through 70 µm cell strainers (Greiner Bio-One, Milan, Italy). Adult liver leukocytes were then separated by centrifugation through 40% Percoll. Peritoneal cavity was washed by 5 mL of PBS and the wash fluid was extracted using a syringe and a 21 G needle (Terumo Corporation, Tokyo, Japan). BM cells were obtained by flushing out the marrow fraction of femurs and tibias. To collect salivary gland cells, submandibular and sublingual glands were minced with scissors and incubated at 37°C for 1 hr in RPMI 1640 medium containing 10% fetal bovine serum, 1 mg/mL collagenase D, and 50 µg/mL DNase I (Sigma-Aldrich, St. Luis, MO, USA). The cell suspension was filtered through a 70 µm cell strainer and purified using 40% Percoll. For the isolation of intestinal lamina propria lymphocytes, small intestines were flushed out and Peyer's patches were excised. The intestines were opened longitudinally, cut into 1 cm pieces, and incubated at 37°C for 30 min in PBS with 5 mM EDTA to remove epithelial cells. The incubated pieces were then minced and digested by RPMI 1640 medium containing 10% fetal bovine serum, 1.25 mg/mL collagenase D, and 50 µg/mL DNase I. The tissue suspension was passed through a 70 µm cell strainer and lymphocytes were purified by 40% Percoll.

## Flow cytometry and cell sorting

Following fluorescent dye- or biotin-conjugated antibodies (BioLegend, San Diego, CA, USA; Thermo Fisher Scientific, Waltham, MA, USA; BD Bioscience, San Jose, CA, USA; TONBO Biosciences, San Diego, CA, USA) were used: CD3ε (145–2 C11), NK1.1 (PK136), NKp46 (29A1.4), CD49a (HMα1), CD49b (DX5), IL-7Rα (A7R34), CXCR6 (SA051D1), TRAIL (N2B2), CD69 (H1.2F3), CD200R (OX-110), CXCR3 (CXCR3-173), CD25 (PC61), Thy-1.2 (30-H12), KLRG1 (2F1/KLRG1), CD11b (M1/70), CD62L (MEL-14), Eomes (Dan11mag), T-bet (4B10), CD45.1 (A20), CD45.2 (104), CD45 (30-F11), Ki-67 (SolA15), Bcl-2 (BCL/10C4), CD31 (MEC13.3), LYVE-1 (LVY7), CD122 (TM-β1), Ter119 (Ter119), F4/80 (BM8), Gr-1 (RB6-8C5), CD19 (6D5), B220 (RA3-6B2), TCRb (H57-597), FcεRI (MAR-1), PD-1 (29 F.1A12), α4β7 (DATK32), Sca-1 (E13-161.7), c-Kit (2B8), Flt3 (A2F10), granzyme B (NGZB), granzyme C (SFC1D8), FasL (MFL3), Ly49E/F (CM4), p-S6 (D57.2.2E), p-STAT5 (47), and p-Akt (S473) (M89-61). Biotinylated monoclonal antibodies were detected with APC- or Brilliant Violet 421-conjugated streptavidin (Thermo Fisher Scientific). For intracellular staining of Eomes, T-bet, Bcl-2, Ki-67, and granzymes, cells were stained for surface antigens, fixed, permeabilized, and stained using Foxp3 Staining Buffer Set or IC Fixation Buffer (Thermo Fisher Scientific). For intracellular staining of p-S6, p-STAT5, and p-Akt (S473), cells were stained for surface antigens, fixed, permeabilized, and stained using BD Phosflow Buffer (BD Biosciences). Flow cytometry and cell sorting were performed on BD FACSVerse or BD LSRFortessa X-20 flow cytometers (BD Biosciences) and BD FACS Aria Ⅱ or Aria Ⅲ cell sorters (BD Biosciences), respectively. Data were analyzed on FlowJo software (FlowJo, Ashland, OR, USA). Debris and dead cells were excluded from analysis by forward and side scatter and propidium iodide (PI) gating. In figures, values in quadrants, gated areas, and interval gates indicate percentages in each population.

## Fate-mapping experiment

Fate-mapping of ILCPs in BM or FL were performed as described previously (*Constantinides et al., 2014*). In brief, 1×10⁴ YFP⁻Lin⁻Sca1⁺c-Kit⁺ (LSK) cells isolated from BM or FL of PLZF<sup>GFPcre/+</sup> Rosa26-YFP

mice were injected i.v. into lethally (9 Gy) irradiated CD45.1 WT mice to remove the random YFP labelling occurred prior to the hematopoiesis. The recipient mice were analyzed 5 weeks after the transplantation. Straight PLZF$^{GFPcre/+}$ Rosa26-YFP mice were also analyzed to confirm the results. For fate-mapping of FL G1-ILCs in Ncr1-CreERT2 Rosa26-tdTomato mice, 4 mg tamoxifen (Sigma-Aldrich) was intraperitoneally injected into pregnant mothers at E17.5. Neonatal (postnatal day 0–4) and 4 weeks old pups were analyzed.

## RNA sequencing (RNA-seq) and data analysis

For bulk RNA-seq, freshly sorted G1-ILC populations ($1\times10^3$ cells) were lysed with Buffer RLT (Qiagen, Hilden, Germany) and purified with RNAClean XP (Beckman Coulter, Brea, CA, USA). Double strand cDNA was synthesized, and sequencing libraries were constructed using SMART-seq HT Plus kit (Takara Bio, Otsu, Japan). Sequencing was performed with 150 bp paired-end reads on the Illumina HiSeq X sequencer (Illumina, San Diego, CA, USA). fastp (*Chen et al., 2018*) was used to assess sequencing quality and to exclude low-quality reads and adaptor contaminations. Reads were mapped on the mouse reference genome (mm10) using HiSat2. The read counts were determined at the gene level with featureCounts. Normalization of gene expression levels and differential gene expression analysis were performed using DESeq2. Genes were considered as differentially expressed genes (DEG) when they had an adjusted p ($p_{adj}$) value <0.05 and fold changes >1.0. Metascape (*Zhou et al., 2019*) and gene set enrichment analysis (GSEA, Broad Institute) was used for enrichment analysis. For reanalysis of single nuclei RNA-seq (snRNA-seq) data of whole liver cells in mice (Liver Cell Atlas; https://www.livercellatlas.org/), normalization, scaling, and UMAP clustering using first 5 dimensions in principal component analysis (PCA) of scaled count matrix were performed on R package Seurat 4.0.2.

## In vitro stimulation

Bulk liver leukocytes ($1\times10^6$ cells) were cultured with RPMI 1640 medium containing 10% FBS, 50 μM 2-mercaptethanol, and 10 mM HEPES (pH7.4). For in vitro stimulation of G1-ILCs, cells were cultured with either 100 ng/mL IL-15, 50 ng/mL IL-12 and IL-18, 20 μg/mL plate-bound antibody for NK1.1 (PK136), NKp46 (29A1.4), NKG2D (CX5), or 2B4 (m2B4 (B6)458.1) (all purchased from BioLegend), or $5\times10^5$ Hepa1-6 cells. After 8 hr, all cells were stained with relevant antibodies for flow cytometry.

## In vivo treatment

For in vivo stimulation of G1-ILCs, mice were administrated intraperitoneally (i.p.) with 2 μg IL-15/IL-15Rα complex (the RLI form as in *Mortier et al., 2008*, provided by Dr. J. M. Dijkstra) once a day. After 18 hr, liver cells were isolated and analyzed by flow cytometry to detect cytotoxic molecule expression. For rapamycin treatment, 30 μg rapamycin in 100 μL corn oil was injected i.p. into mice 18 hr before the analysis.

## Intrasplenic injection

A small incision was made on the left flank of anesthetized mice and the lower pole of the spleen was gently exposed. Cell suspension (50 μL) was slowly injected into the spleen by a 0.3 mL insulin syringe with a 29 G needle (BD Biosciences). Cotton wool was applied to the spleen for several minutes after the injection to stop bleeding.

## Adoptive transfer experiments

NK cells, 7 R$^−$ ILC1s, 7R$^+$ ILC1s, and BM iILC1s ($2\times10^4$–$5\times10^4$ cells) and FL G1-ILCs ($1\times10^5$ cells) were sorted from adult and E18.5 CD45. 1 WT mice, respectively. Each G1-ILC population was adoptively transferred into CD45.2 WT mice by intrasplenic injection. At the indicated time points, cells were isolated from the liver of recipient mice and analyzed by flow cytometry. To assess the stability and the proliferation capacity of 7 R$^−$ ILC1s and 7R$^+$ ILC1s in inflammatory conditions, CD45.2 WT mice were injected with indicated G1-ILC populations labeled by Cell Proliferation Dye (CPD) eFluor 450 (Thermo Fisher Scientific) (day 0) followed by the administration of 2 μg IL-15/IL-15Rα at days 1, 3, and 5. After 7 days, liver leukocytes of recipient mice were analyzed.

## Immunofluorescence

Whole FL of WT mice or hepatic lobes of adult *Cxcr6*$^{GFP/+}$ mice were fixed with 4% paraformaldehyde (PFA) at 4°C for 6 hr, embedded in Optimal Cutting Temperature compound (Sakura Finetechnical,

Tokyo, Japan), and sliced with a cryomicrotome (Leica CM3050S, Wetzlar, Germany). Tissue sections were stained at room temperature for 1 hr with primary antibodies as follows: biotin-anti-CD31, rabbit polyclonal IgG anti-LYVE-1 (RELIATech, Braunschweig, Germany), and goat polyclonal IgG anti-NKp46 (R&D, Minneapolis, MN, USA). Following secondary antibodies (BioLegend) were used: Dylight 488 anti-rabbit IgG, Alexa Fluor 555 anti-rabbit IgG, Alexa Fluor 647 anti-goat IgG, Brilliant Violet 421 anti-rabbit IgG, Dylight 649 anti-rat IgG, and PE anti-rat IgG. Biotinylated antibodies were detected with FITC-, PE-, or Brilliant Violet 421-conjugated streptavidin (Thermo Fisher Scientific). Stained sections were then mounted using PermaFluor Aqueous Mounting Medium (Thermo Fisher Scientific) and examined by a TCS SP8 confocal microscope (Leica) with HC PLAPO CS2 20×/0.75 IMM or HC PLAPO CS2 40×/1.30 OIL object lenses. Multiple image stacks (10–20 µm) were acquired with two times of frame averaging and maximum projection was then performed on using LAS X software (Leica).

## Intravascular staining

PE/Cy7 anti-CD45.2 antibody (clone: 104) (2 µg) was injected intravenously (i.v.) into mice 2 min before the perfusion and liver dissection. Isolated liver leukocytes were stained with fluorophore-conjugated antibodies of interest in addition to APC/Cy7 anti-CD45 antibody (clone: 30-F11) and analyzed by flow cytometry.

## Parabiosis

Female CD45.1 and CD45.2 congenic C57BL/6 J mice were surgically conjoined as previously described (*Gasteiger et al., 2015*). In brief, lateral skin from elbow to knee of each mouse was sutured, forelimbs and hindlimbs were tied together, and the skin incisions were closed using surgical adhesive. After 60 days of surgery, mice were analyzed by flow cytometry.

## Cell lines

Hepa1-6 cells were purchased from RIKEN BioResource Center (RIKEN BRC, Tsukuba, Japan), which is authenticated by STR profiling (https://www.cellosaurus.org/CVCL_0327). B16F10 cells were provided by Dr. T. Honjo at Kyoto University. Both cell lines were confirmed to be mycoplasma negative before use, and were maintained in Dulbecco's modified Eagle's medium (DMEM) supplemented with 10% FBS, 2 mM L-glutamine, and antibiotics.

## In vitro killing assay

For time-lapse killing assays, $2 \times 10^2$ Hepa1-6 cells or B16F10 cells were labeled by CPD eFluor 450 and pre-incubated with RPMI 1640 medium (without phenol red) containing 10% FBS, 10 mM HEPES (pH7.4), antibiotics, and 1 µg/mL PI on 96-well round bottom plates. Freshly sorted liver NK cells, 7 R$^-$ ILC1s, or 7R$^+$ ILC1s ($2 \times 10^3$ cells) were added to the plates and co-cultured with tumor cells. Time-lapse imaging was performed using a BZ-X710 microscope (Keyence, Osaka, Japan) with CFI Plan Apo $\lambda$ 10×and CFI Plan Fluor DL 10×objective lenses at a 20 min interval for up to 6 hr. To determine the contribution of effector molecules, 50 nM concanamycin A (CMA), 10 µg/mL anti-TRAIL antibody (N2B2), or 10 µg/mL anti-FasL antibody (MFL3) were supplemented and compared to vehicle-supplemented controls. Tumor cell viability was defined by the ratio of the CPD$^+$PI$^-$ viable tumor cell number at each time point to the viable tumor cell number at the beginning of the imaging and represented as the moving average of three consecutive time points. Image analysis and cell counts were performed using BZ-X Analyzer (Keyence). For killing assay of YAC-1 cells (provided by Dr. M. Hattori at Kyoto University), $2 \times 10^4$ freshly sorted liver NK cells, 7 R$^-$ ILC1s, or 7R$^+$ ILC1s were co-cultured for 4 hr with $2 \times 10^3$ CPD eFluor 450-labeled YAC-1 cells in RPMI 1640 medium containing 10% FBS, 10 mM HEPES (pH7.4), and antibiotics. After culture, YAC-1 cells were stained with FITC-conjugated Annexin V and PI (MEBCYTO-Apoptosis Kit, MBL) and the ratio of Annexin V$^+$PI$^+$ apoptotic cells were analyzed by flow cytometry.

## Statistical analysis

Statistical differences were evaluated by the two-tailed unpaired Student's *t*-test and one-way or two-way analysis of variance (ANOVA) using GraphPad Prism 8 (GraphPad Software, San Diego, California, USA). Asterisks in all figures indicate as follows: *$p<0.05$, **$p<0.01$, ***$p<0.001$, and ****$p<0.0001$.

## Acknowledgements

We acknowledge Dr. Lewis L Lanier for providing Ncr1-CreERT2 mice, Dr. Mark A Magnuson for Alb-Cre mice, Dr. Jason Cyster for *Lyve1*^Cre mice, and members of the K Ikuta laboratory for discussion and supervision. Time-lapse imaging using Keyence BZ-X710 microscope were performed at the Medical Research Support Center, Graduate School of Medicine, Kyoto University, which was supported by Platform for Drug Discovery, Informatics, and Structural Life Science from the Ministry of Education, Culture, Sports, Science and Technology, Japan.

Funding: This research was supported by the Japan Society for the Promotion of Science (JSPS) KAKENHI grant numbers 20H03501 and 20K21525 (K.I.), 19K16687 and 21K07067 (G.C.). It is also supported by a grant from Takeda Science Foundation to A.S., by grants from the Shimizu Foundation for Immunology and Neuroscience to T.A. and A.S., and by the Joint Usage/Research Center program of Institute for Life and Medical Sciences Kyoto University.

## Additional information

### Funding

| Funder | Grant reference number | Author |
|---|---|---|
| Japan Society for the Promotion of Science | 20H03501 | Koichi Ikuta |
| Japan Society for the Promotion of Science | 20K21525 | Koichi Ikuta |
| Japan Society for the Promotion of Science | 19K16687 | Guangwei Cui |
| Japan Society for the Promotion of Science | 21K07067 | Guangwei Cui |
| Takeda Science Foundation | | Akihiro Shimba |
| Shimizu Foundation for Immunology and Neuroscience | | Takuma Asahi |

The funders had no role in study design, data collection and interpretation, or the decision to submit the work for publication.

### Author contributions

Takuma Asahi, Conceptualization, Data curation, Formal analysis, Funding acquisition, Writing – original draft, Project administration; Shinya Abe, Keizo Ohira, Formal analysis, Investigation, Writing – review and editing; Guangwei Cui, Akihiro Shimba, Formal analysis, Funding acquisition, Investigation, Writing – review and editing; Tsukasa Nabekura, Johannes M Dijkstra, Masaki Miyazaki, Akira Shibuya, Resources, Supervision, Writing – review and editing; Hitoshi Miyachi, Satsuki Kitano, Methodology, Writing – review and editing; Hiroshi Ohno, Resources, Writing – review and editing; Koichi Ikuta, Conceptualization, Supervision, Funding acquisition, Writing – original draft, Project administration, Writing – review and editing

### Author ORCIDs

Takuma Asahi http://orcid.org/0000-0001-6372-7563
Johannes M Dijkstra http://orcid.org/0000-0001-7097-3826
Akira Shibuya http://orcid.org/0000-0002-4480-4858
Koichi Ikuta http://orcid.org/0000-0003-1319-1021

### Ethics

All protocols of animal experiments were approved by the Animal Experimentation Committee of the Institute for Life and Medical Sciences, Kyoto University (reference number for approval: A20-2-5, A21-3-2, A22-4).

Decision letter and Author response
Decision letter https://doi.org/10.7554/eLife.84209.sa1
Author response https://doi.org/10.7554/eLife.84209.sa2

## Additional files

### Supplementary files
• MDAR checklist

### Data availability
Sequencing data generated in our RNA-seq experiments have been deposited in Gene Expression Omnibus (GEO) under accession code GSE205894.

The following dataset was generated:

| Author(s) | Year | Dataset title | Dataset URL | Database and Identifier |
|---|---|---|---|---|
| Ikuta K, Asahi T | 2022 | Lineage heterogeneity underlies the multiple origins and functions of type 1 innate lymphoid cells | https://www.ncbi.nlm.nih.gov/geo/query/acc.cgi?acc=GSE205894 | NCBI Gene Expression Omnibus, GSE205894 |

The following previously published dataset was used:

| Author(s) | Year | Dataset title | Dataset URL | Database and Identifier |
|---|---|---|---|---|
| Guilliams M, Bonnardel J, Haest B, Vanderborght B | 2022 | Spatial proteogenomics reveals distinct and evolutionarily-conserved hepatic macrophage niches | https://www.ncbi.nlm.nih.gov/geo/query/acc.cgi?acc=GSE192742 | NCBI Gene Expression Omnibus, GSE192742 |

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
