## [Editor Report]

This study provides important insights into the developmental process and functional heterogeneity of liver ILC1s, especially how IL-7R+ and IL-7R- ILC1s are generated. The authors present compelling evidence on the dependence of ILC1s on IL-7R- precursor and their reliance on IL-15 to develop cytotoxic functions. The work will be of broad interest to immunologists and liver biologists.

---

## [Decision Letter]

**Decision letter after peer review:**

Thank you for submitting your article "Liver type 1 innate lymphoid cells lacking IL-7 receptor are a native killer cell subset fostered by parenchymal niches" for consideration by *eLife*. Your article has been reviewed by 2 peer reviewers, and the evaluation has been overseen by a Reviewing Editor and Satyajit Rath as the Senior Editor. The reviewers have opted to remain anonymous.

Essential revisions:

Please address the clarifications as well as corrections highlighted by both reviewers, especially the point made by reviewer 1 on "how many experiments and mice were analyzed in each experimental set".

*Reviewer #1 (Recommendations for the authors):*

General comments:

– L. 60-62: both of these progenitors are not restricted to ILC1s and can also give rise to NK cells (though to a lesser extent).

– L. 180: Song et al. use Ncr1Cre or VavCre Rorafl/fl mice and not full knock-out mice, this should be clarified in the text.

– L. 194: Please indicate how many times the IL-15/IL-15Rα complex was injected (rather than just say repeatedly), either in the text of the figure legend.

– L. 194: What is CPD?

– Figure 2J: Title does not match up with y-axis (title says number and y axis freq. of cells). What is the grey bar?

– Figure 2 H/I/K: What are the ~10% of cells that do convert? It could be useful to include other markers that are co-expressed with CD127 according to Friedrich et al. (for example IL-18R) to determine whether the cells that lose IL-7R expression also lose expression of other markers or just downregulate IL-7R and vice versa.

– Figure 4B: change the label to CXCR6-GFP.

– Supplementary Figure 5A shows that not all ILC1s express CXCR6 – what is the different between the CXCR6+ and – cells – do they express the same levels of IL-7R? This is important for the conclusions on the imaging using CXCR6-GFP mice.

– Figure 6: it would be good to show representative FACS plots of GzB and FasL expression to show what percentage of cells expresses these molecules.

– Figure 6D: Please add the legend to this panel.

– PLZF-fm has been used – what is the evidence that only ILC1 express PLZF?

– IL-15/IL-15Rα has been used as a model of an "inflammatory state" – could responsiveness be explained by IL-15R expression?

Figures: The figures are well presented. While it is evident that multiple experiments were performed, it is not completely clear how many experiments and mice were analyzed in each experimental set. Please clarify this throughout the figures.

*Reviewer #2 (Recommendations for the authors):*

More detailed investigation into these claims would be welcomed given ILC1's are known to express different effector molecules (eg. Granzyme C and TRAIL) compared to NK cells.

Questions of interest would be in vitro analysis of ILC1 v NK cell effector responses in response to cytokine, tumor cells, antibody restimulation against activating receptors such as NK1.1, NKp46, NKG2D, 2B4, Ly49H

The big unresolved question is why ILC1 dominate fetal innate lymphocytes versus NK cells in adult life. Are ILC1 more regulatory versus NK cells being more pro-inflammatory effectors?

---

## [Author Response]

Essential revisions:Reviewer #1 (Recommendations for the authors):General comments:– L. 60-62: both of these progenitors are not restricted to ILC1s and can also give rise to NK cells (though to a lesser extent).

As pointed out by the reviewer, ILCPs and LSM cells can also generate NK cells, though they preferentially differentiate into helper ILCs including ILC1s. Following the suggestion, we have amended the text on page 3, lines 58-62.

– L. 180: Song et al. use Ncr1Cre or VavCre Rorafl/fl mice and not full knock-out mice, this should be clarified in the text.

We appreciate the reviewer for pointing out the insufficient information on previous study. As suggested by the reviewer, we have amended the text on page 6-7, lines 182-183.

– L. 194: Please indicate how many times the IL-15/IL-15Rα complex was injected (rather than just say repeatedly), either in the text of the figure legend.

Following the suggestion of the reviewer, we have amended the figure legend text on page 29, lines 878-880.

– L. 194: What is CPD?

We apologize that we did not explain the abbreviation. To assess how many times transferred cells proliferate in the host mice, we stained sorted cells by Cell Proliferation Dye (CPD) eFluor 450 (Thermo Fisher Scientific), transferred them into hosts, and calculated the number of cell divisions based on the flow cytometric analysis of the dye dilution. We have amended the text on page 7, lines 198.

– Figure 2J: Title does not match up with y-axis (title says number and y axis freq. of cells). What is the grey bar?

Figure 2J shows the number of divisions of transferred cells (horizontal axis) in response to IL-15/IL-15Ra administration. Grey, blue, and orange bars indicate NK cells, 7R^−^ ILC1s, and 7R^+^ ILC1s, respectively. Cells shaped 5 or more populations based on the CPD dye dilution, which corresponds to how many times they proliferated. We then expressed the “frequency” of each cell population as a vertical axis. We apologize that the description was confusing. To make this point clear, we have added the annotation to Figure 2J and amended the figure legend text on page 29, lines 879-880.

– Figure 2 H/I/K: What are the ~10% of cells that do convert? It could be useful to include other markers that are co-expressed with CD127 according to Friedrich et al. (for example IL-18R) to determine whether the cells that lose IL-7R expression also lose expression of other markers or just downregulate IL-7R and vice versa.

Following the suggestion of the reviewer, we analyzed IL-7R and IL-18R1 expression on adoptively transferred 7R^−^ and 7R^+^ ILC1s. Although the expression of IL-18R1 (and another marker such as CD3g, CD3d, or Kit) alone could not completely distinguish two ILC1s (Friedrich et al., 2021), co-staining of IL-7R and IL-18R1 made our FACS gating clear. We found that 7R^+^ ILC1s maintained high expression of IL-18R1 as well as IL-7R for 4 weeks. The downregulation of IL-7R was not obviously correlated to that of IL-18R1. Because IL-7R may be the best, but not a perfect marker for distinguishing two ILC1s, ~10% conversion might be due to the contamination during FACS sorting or analysis. We have added these results in Figure 2—figure supplement 1A and amended the text on page 7, lines 190-191 and page 33, lines 1007-1009.

– Figure 4B: change the label to CXCR6-GFP.

Following the suggestion of the reviewer, we changed the label of Figure 4B.

– Supplementary Figure 5A shows that not all ILC1s express CXCR6 – what is the different between the CXCR6+ and – cells – do they express the same levels of IL-7R? This is important for the conclusions on the imaging using CXCR6-GFP mice.

Following the suggestion of the reviewer, we analyzed GFP expression on 7R^−^ and 7R^+^ ILC1s in CXCR6-GFP mice and found that they expressed GFP at similar levels (about 70% are GFP^+^). We have added these results in Figure 5—figure supplement 1A.

– Figure 6: it would be good to show representative FACS plots of GzB and FasL expression to show what percentage of cells expresses these molecules.

As suggested by the reviewer, we have added the representative FACS profiles of granzyme B expression in Figure 6A, and FasL and TRAIL expression in Figure 6—figure supplement 1A and 1B, respectively.

– Figure 6D: Please add the legend to this panel.

As suggested by the reviewer, we added the legend to Figure 6E (originally Figure 6D).

– PLZF-fm has been used – what is the evidence that only ILC1 express PLZF?

We apologize for the confusion. We used the PLZF-fm system which was previously reported in Nature (Constantinides et al., 2014). In brief, taking advantage of the transient PLZF expression at the ILCP stage, but NOT at mature ILCs including ILC1s and NK cells, we can trace ILCP progenies in PLZF-Cre Rosa26-reporter mice. We utilized these mice in order to ask whether FL G1-ILCs and 7R^−^ ILC1s are actually derived from ILCPs and different from the NK lineage. Thus, ILC1s do not express PLZF and that is not a problem. To make this point clear, we have amended the text on page 6, lines 153-158.

– IL-15/IL-15Rα has been used as a model of an "inflammatory state" – could responsiveness be explained by IL-15R expression?

Following the suggestion of the reviewer, we analyzed the IL-15Ra expression on liver G1-ILC subsets. As shown in Author response image 1, we could not detect IL-15Ra expression in NK cells, 7R^−^ ILC1s, and 7R^+^ ILC1s by flow cytometry. The PE-conjugated antibody we used for IL-15Ra staining seemed to work, because a small PE^+^ population was observed within non-lymphocytes (lower FACS plots). In addition, gene expression of IL-15Ra and common g chain (g_c_) were similar among three G1-ILC subsets in RNA-seq (data not shown). Thus, the expression of IL-15Ra and g_c_ may not explain the heterogenous responsiveness in G1-ILCs. In contrast, gene expression of IL-2Rb, a β subunit of IL-15R, is higher in NK cells than 7R^−^ and 7R^+^ ILC1s (data not shown). Therefore, it is possible that IL-2Rb expression accounts for the NK-ILC1 difference, but the difference between 7R^−^ and 7R^+^ ILC1s still could not be explained by the expression of IL-15R components.

**Author response image 1. sa2fig1:** 

Figures: The figures are well presented. While it is evident that multiple experiments were performed, it is not completely clear how many experiments and mice were analyzed in each experimental set. Please clarify this throughout the figures.

Following the suggestion of the reviewer, we have amended the figure legend text throughout the figures (because almost all figure legends are corrected, they are not highlighted in detail). We have also provided source data for figures including summary data (bar graphs, heatmaps, and line graphs).

Reviewer #2 (Recommendations for the authors):More detailed investigation into these claims would be welcomed given ILC1's are known to express different effector molecules (eg. Granzyme C and TRAIL) compared to NK cells.

As pointed out by the reviewer, ILC1s express granzyme C and TRAIL in addition to granzyme B. To address the expression of these molecules on ILC1s under steady- and activated states, we performed in vitro culture experiments. Further details are given in the next section.

Questions of interest would be in vitro analysis of ILC1 v NK cell effector responses in response to cytokine, tumor cells, antibody restimulation against activating receptors such as NK1.1, NKp46, NKG2D, 2B4, Ly49H

Following the suggestion of the reviewer, we assessed granzyme B, TRAIL, and granzyme C expression on G1-ILCs in response to in vitro exposure of cytokines (IL-15 and IL-12/IL-18), cross-linking of NK receptors (NK1.1, NKp46, NKG2D, and 2B4), and coculture with Hepa1-6 cells. IL-15 enhanced granzyme B expression on whole G1-ILC subsets, but NK cell-expressed granzyme B was the most responsive, consistent with in vivo stimulation with IL-15/IL-15Ra injection. In contrast, TRAIL and granzyme C expression of ILC1s, but not NK cells, highly and variably responded to each type of stimulus. We have added these results in Figure 6—figure supplement 1E and amended the text on page 10, lines 297-303.

The big unresolved question is why ILC1 dominate fetal innate lymphocytes versus NK cells in adult life. Are ILC1 more regulatory versus NK cells being more pro-inflammatory effectors?

We appreciate the reviewer for the pivotal question. As is the case with ILC2s and ILC3s, non-inflammatory roles of ILC1s have also been reported. For example, ILC1s exert a tolerogenic effect via T cells in the liver (Zhou et al., 2019) and are involved in angiogenesis and tissue remodeling in specific conditions such as tumor microenvironment (Sivori et al., 2020) and uterine remodeling (Murphy et al., 2022). Whether FL G1-ILCs and 7R^−^ ILC1s have such a non-inflammatory or physiological role is actually unknown, because they express great amounts of cytotoxic granules but hardly express anti-inflammatory cytokines, ligands, and growth factors. However, it is possible that FL G1-ILCs and 7R^−^ ILC1s contribute to early liver development by eliminating unnecessary cells. We have discussed this on page 13, lines 397-400.

References

Constantinides, M.G., B.D. McDonald, P.A. Verhoef, and A. Bendelac. 2014. A committed precursor to innate lymphoid cells. *Nature* 508:397-401.

Friedrich, C., R.L.R.E. Taggenbrock, R. Doucet-Ladevèze, G. Golda, R. Moenius, P. Arampatzi, N.A.M. Kragten, K. Kreymborg, M. Gomez De Agüero, W. Kastenmüller, A.-E. Saliba, D. Grün, K.P.J.M. Van Gisbergen, and G. Gasteiger. 2021. Effector differentiation downstream of lineage commitment in ILC1s is driven by Hobit across tissues. *Nat. Immunol.* 22:1256-1267.

Murphy, J.M., L. Ngai, A. Mortha, and S.Q. Crome. 2022. Tissue-dependent adaptations and functions of innate lymphoid cells. *Front. Immunol.* 13:836999.

Sivori, S., D. Pende, L. Quatrini, G. Pietra, M. Della Chiesa, P. Vacca, N. Tumino, F. Moretta, M.C. Mingari, F. Locatelli, and L. Moretta. 2020. NK cells and ILCs in tumor immunotherapy. *Mol. Aspects. Med.* 80:100870.

Zhou, J., H. Peng, K. Li, K. Qu, B. Wang, Y. Wu, L. Ye, Z. Dong, H. Wei, R. Sun, and Z. Tian. 2019. Liver-resident NK cells control antiviral activity of hepatic T cells via the PD-1-PD-L1 axis. *Immunity* 50:403-417.e404.